# Low-Rank Tucker Decomposition of Large Tensors Using TensorSketch

**Osman Asif Malik**
Department of Applied Mathematics
University of Colorado Boulder
`osman.malik@colorado.edu`

**Stephen Becker**
Department of Applied Mathematics
University of Colorado Boulder
`stephen.becker@colorado.edu`

## Abstract

We propose two randomized algorithms for low-rank Tucker decomposition of tensors. The algorithms, which incorporate sketching, only require a single pass of the input tensor and can handle tensors whose elements are streamed in any order. To the best of our knowledge, ours are the only algorithms which can do this. We test our algorithms on sparse synthetic data and compare them to multiple other methods. We also apply one of our algorithms to a real dense 38 GB tensor representing a video and use the resulting decomposition to correctly classify frames containing disturbances.

## 1  Introduction

Many real datasets have more than two dimensions and are therefore better represented using tensors, or multi-way arrays, rather than matrices. In the same way that methods such as the singular value decomposition (SVD) can help in the analysis of data in matrix form, tensor decompositions are important tools when working with tensor data. As multidimensional datasets grow larger and larger, there is an increasing need for methods that can handle them, even on modest hardware. One approach to the challenge of handling big data, which has proven to be very fruitful in the past, is the use of randomization. In this paper, we present two algorithms for computing the Tucker decomposition of a tensor which incorporate random sketching. A key challenge to incorporating sketching in the Tucker decomposition is that the relevant design matrices are Kronecker products of the factor matrices. This makes them too large to form and store in RAM, which prohibits the application of standard sketching techniques. Recent work [26, 27, 2, 10] has led to a new technique called TENSORSKETCH which is ideally suited for sketching Kronecker products. It is based on this technique that we develop our algorithms. Our algorithms, which are single pass and can handle streamed data, are suitable when the decomposition we seek is of low-rank. When we say that our algorithms can handle streamed data, we mean that they can decompose a tensor whose elements are revealed one at a time and then discarded, no matter which order this is done in. These streaming properties of our methods follow directly from the streaming properties of TENSORSKETCH.

In some applications, such as the compression of scientific data produced by high-fidelity simulations, the data tensors can be very large (see e.g. the recent work [1]). Since such data frequently is produced incrementally, e.g. by stepping forward in time, a compression algorithm which is one-pass and can handle the tensor elements being streamed would make it possible to compress the data without ever having to store it in full. Our algorithms have these properties.

In summary, our paper makes the following algorithmic contributions:

- We propose two algorithms for Tucker decomposition which incorporate TENSORSKETCH. They are intended to be used for low-rank decompositions.

- We propose an idea for defining the sketch operators upfront. In addition to increasing accuracy and reducing run time, it allows us to make several other improvements. These include only requiring a single pass of the data, and being able to handle tensors whose elements are streamed. To the best of our knowledge, ours are the only algorithms which can do this.

## 1.1 A brief introduction to tensors and the Tucker decomposition

We use the same notations and definitions as in the review paper by Kolda and Bader [18]. Due to limited space, we only explain our notation here, with definitions given in Section S1 of the supplementary material. A *tensor* $\mathcal{X} \in \mathbb{R}^{I_1 \times I_2 \times \cdots \times I_N}$ is an array of dimension $N$, also called an $N$-way tensor. Boldface Euler script letters, e.g. $\mathcal{X}$, denote tensors of dimension 3 or greater; bold capital letters, e.g. $\mathbf{X}$, denote matrices; bold lowercase letters, e.g. $\mathbf{x}$, denote vectors; and lowercase letters, e.g. $x$, denote scalars. For scalars indicating dimension size, uppercase letters, e.g. $I$, will be used. "$\otimes$" and "$\odot$" denote the *Kronecker* and *Khatri-Rao products*, respectively. The *mode-n matricization* of a tensor $\mathcal{X} \in \mathbb{R}^{I_1 \times I_2 \times \cdots \times I_N}$ is denoted by $\mathbf{X}_{(n)} \in \mathbb{R}^{I_n \times \prod_{i \neq n} I_n}$. Similarly, $\mathbf{x}_{(:)} \in \mathbb{R}^{\prod_n I_n}$ denotes the vectorization of $\mathcal{X}$ into a column vector. The *n-mode tensor-times-matrix (TTM) product* of $\mathcal{X}$ and a matrix $\mathbf{A} \in \mathbb{R}^{J \times I_n}$ is denoted by $\mathcal{X} \times_n \mathbf{A} \in \mathbb{R}^{I_1 \times \cdots \times I_{n-1} \times J \times I_{n+1} \times \cdots \times I_N}$. The *norm* of $\mathcal{X}$ is defined as $\|\mathcal{X}\| = \|\mathbf{x}_{(:)}\|_2$. For a positive integer $n$, we use the notation $[n] := \{1, 2, \ldots, n\}$.

There are multiple tensor decompositions. In this paper, we consider the *Tucker decomposition*. A Tucker decomposition of a tensor $\mathcal{X} \in \mathbb{R}^{I_1 \times I_2 \times \cdots \times I_N}$ is

$$\mathcal{X} = \mathcal{G} \times_1 \mathbf{A}^{(1)} \times_2 \mathbf{A}^{(2)} \cdots \times_N \mathbf{A}^{(N)} =: [\![\mathcal{G}; \mathbf{A}^{(1)}, \mathbf{A}^{(2)}, \ldots, \mathbf{A}^{(N)}]\!], \tag{1}$$

where $\mathcal{G} \in \mathbb{R}^{R_1 \times R_2 \times \cdots \times R_N}$ is called the *core tensor* and each $\mathbf{A}^{(n)} \in \mathbf{R}^{I_n \times R_n}$ is called a *factor matrix*. Without loss of generality, the factor matrices can be assumed to have orthonormal columns, which we will assume as well. We say that $\mathcal{X}$ in (1) is a rank-$(R_1, R_2, \ldots, R_N)$ tensor.

The Tucker decomposition problem of decomposing a data tensor $\mathcal{Y} \in \mathbb{R}^{I_1 \times I_2 \times \cdots \times I_N}$ can be formulated as

$$\underset{\mathcal{G}, \mathbf{A}^{(1)}, \ldots, \mathbf{A}^{(N)}}{\arg\min} \left\{ \left\| \mathcal{Y} - [\![\mathcal{G}; \mathbf{A}^{(1)}, \ldots, \mathbf{A}^{(N)}]\!] \right\| : \mathcal{G} \in \mathbb{R}^{R_1 \times \cdots \times R_N}, \mathbf{A}^{(n)} \in \mathbb{R}^{I_n \times R_n} \text{ for } n \in [N] \right\}. \tag{2}$$

The standard approach to this problem is to use alternating least-squares (ALS). By rewriting the objective function appropriately (use e.g. Proposition 3.7 in [17]), we get the following steps, which are repeated until convergence:

1. For $n = 1, \ldots, N$, update $\mathbf{A}^{(n)} = \underset{\mathbf{A} \in \mathbb{R}^{I_n \times R_n}}{\arg\min} \left\| \left( \bigotimes_{\substack{i=N \\ i \neq n}}^{1} \mathbf{A}^{(i)} \right) \mathbf{G}_{(n)}^\top \mathbf{A}^\top - \mathbf{Y}_{(n)}^\top \right\|_F^2. \tag{3}$

2. Update $\mathcal{G} = \underset{\mathcal{Z} \in \mathbb{R}^{R_1 \times \cdots \times R_N}}{\arg\min} \left\| \left( \bigotimes_{i=N}^{1} \mathbf{A}^{(i)} \right) \mathbf{z}_{(:)} - \mathbf{y}_{(:)} \right\|_2^2. \tag{4}$

One can show that the solution for the $n$th factor matrix $\mathbf{A}^{(n)}$ in (3) is given by the $R_n$ leading left singular vectors of the mode-$n$ matricization of $\mathcal{Y} \times_1 \mathbf{A}^{(1)\top} \cdots \times_{n-1} \mathbf{A}^{(n-1)\top} \times_{n+1} \mathbf{A}^{(n+1)\top} \cdots \times_N \mathbf{A}^{(N)\top}$. Since each $\mathbf{A}^{(i)}$ has orthogonal columns, it turns out that the solution to (4) is given by $\mathcal{G} = \mathcal{Y} \times_1 \mathbf{A}^{(1)\top} \times_2 \mathbf{A}^{(2)\top} \cdots \times_N \mathbf{A}^{(N)\top}$. These insights lead to Algorithm 1, which we will refer to as TUCKER-ALS. It is also frequently called *higher-order orthogonal iteration* (HOOI), and is more accurate than *higher-order SVD* (HOSVD) which is another standard algorithm for Tucker decomposition. More details can be found in [18].

## 1.2 A brief introduction to TensorSketch

In this paper, we apply TENSORSKETCH to approximate the solution to large overdetermined least-squares problems, and to approximate chains of TTM products similar to those in (1). TENSORSKETCH is a randomized method which allows us to reduce the cost and memory usage of these computations in exchange for somewhat reduced accuracy. It can be seen as a specialized version of another sketching method called COUNTSKETCH, which was introduced in [7] and further analyzed in [8]. One way to define a COUNTSKETCH operator $\mathbf{S} : \mathbb{R}^I \to \mathbb{R}^J$ is as $\mathbf{S} = \mathbf{PD}$, where

---

**Algorithm 1:** TUCKER-ALS (aka HOOI)

---

**input** : $\mathcal{Y}$, target rank $(R_1, R_2, \ldots, R_N)$
**output** : Rank-$(R_1, R_2, \ldots, R_N)$ Tucker decomposition $[\![\mathcal{G}; \mathbf{A}^{(1)}, \ldots, \mathbf{A}^{(N)}]\!]$ of $\mathcal{Y}$

**1** Initialize $\mathbf{A}^{(2)}, \mathbf{A}^{(3)}, \ldots, \mathbf{A}^{(N)}$
**2 repeat**
**3**     **for** $n = 1, \ldots, N$ **do**
**4**        $\mathcal{Z} = \mathcal{Y} \times_1 \mathbf{A}^{(1)\top} \cdots \times_{n-1} \mathbf{A}^{(n-1)\top} \times_{n+1} \mathbf{A}^{(n+1)\top} \cdots \times_N \mathbf{A}^{(N)\top}$
**5**        $\mathbf{A}^{(n)} = R_n$ leading left singular vectors of $\mathbf{Z}_{(n)}$ /* Solves Eq. (3) */
**6**     **end**
**7 until** *termination criteria met*
**8** $\mathcal{G} = \mathcal{Y} \times_1 \mathbf{A}^{(1)\top} \times_2 \mathbf{A}^{(2)\top} \cdots \times_N \mathbf{A}^{(N)\top}$ /* Solves Eq. (4) */
**9 return** $\mathcal{G}, \mathbf{A}^{(1)}, \ldots, \mathbf{A}^{(N)}$

---

- $\mathbf{P} \in \mathbb{R}^{J \times I}$ is a matrix with $p_{h(i),i} = 1$, and all other entries equal to 0;
- $h : [I] \to [J]$ is a random map such that $(\forall i \in [I])(\forall j \in [J]) \, \mathbb{P}(h(i) = j) = 1/J$; and
- $\mathbf{D} \in \mathbb{R}^{I \times I}$ is a diagonal matrix, with each diagonal entry equal to $+1$ or $-1$ with equal probability.

Due to the special structure of $\mathbf{S}$, it is inefficient to store it as a full matrix. When applying $\mathbf{S}$ to a matrix $\mathbf{A}$, it is better to do this implicitly, which costs only $\mathcal{O}(\mathrm{nnz}(\mathbf{A}))$ and avoids storing $\mathbf{S}$ as a full matrix. Here, $\mathrm{nnz}(\mathbf{A})$ denotes the number of nonzero elements of $\mathbf{A}$.

TENSORSKETCH was first introduced in 2013 in [26] where it is applied to compressed matrix multiplication. In [27], it is used for approximating support vector machine polynomial kernels efficiently. Avron et al. [2] show that TENSORSKETCH provides an oblivious subspace embedding. Diao et al. [10] provide theoretical guarantees which we will rely on in this paper. Below is an informal summary of those results we will use; for further details, see the paper by Diao et al., especially Theorem 3.1 and Lemma B.1. Let $\mathbf{A} \in \mathbb{R}^{L \times M}$ be a matrix, where $L \gg M$. Like other classes of sketches, an instantiation of TENSORSKETCH is a linear map $\mathbf{T} : \mathbb{R}^L \to \mathbb{R}^J$, where $J \ll L$, such that, if $\mathbf{y} \in \mathbb{R}^L$ and $\widetilde{\mathbf{x}} \overset{\text{def}}{=} \arg\min_{\mathbf{x}} \|\mathbf{T}\mathbf{A}\mathbf{x} - \mathbf{T}\mathbf{y}\|_2$, then for $J$ sufficiently large (depending on $\varepsilon > 0$), with high probability $\|\mathbf{A}\widetilde{\mathbf{x}} - \mathbf{y}\|_2 \le (1 + \varepsilon) \min_{\mathbf{x}} \|\mathbf{A}\mathbf{x} - \mathbf{y}\|_2$.

The distinguishing feature of TENSORSKETCH is that if the matrix $\mathbf{A}$ is of the form $\mathbf{A} = \mathbf{A}^{(N)} \otimes \mathbf{A}^{(N-1)} \otimes \cdots \otimes \mathbf{A}^{(1)}$, where each $\mathbf{A}^{(n)} \in \mathbb{R}^{I \times R}$, $I \gg R$, then the cost of computing $\mathbf{T}\mathbf{A}$ can be shown to be $\mathcal{O}(NIR + JR^N)$, excluding log factors, whereas naïve matrix multiplication would cost $\mathcal{O}(JI^N R^N)$. Moreover, $\mathbf{T}\mathbf{A}$ can be computed without ever forming the full matrix $\mathbf{A}$. One can show that this is achievable by first applying an independent COUNTSKETCH operator $\mathbf{S}^{(n)} \in \mathbb{R}^{J \times I}$ to each factor matrix $\mathbf{A}^{(n)}$ and then computing the full TENSORSKETCH using the fast Fourier transform (FFT). The formula for this is

$$\mathbf{T}\mathbf{A} = \mathbf{T} \bigotimes_{i=N}^{1} \mathbf{A}^{(i)} = \mathrm{FFT}^{-1} \left( \left( \bigodot_{i=N}^{1} \left( \mathrm{FFT} \left( \mathbf{S}^{(i)} \mathbf{A}^{(i)} \right) \right)^{\top} \right)^{\top} \right). \tag{5}$$

These results generalize to the case when the factor matrices are of different sizes. In Section S2 of the supplementary material, we provide a more thorough introduction to COUNTSKETCH and TENSORSKETCH, including how to arrive at the formula (5).

## 2 Related work

Randomized algorithms have been applied to tensor decompositions before. Wang et al. [31] and Battaglino et al. [5] apply sketching techniques to the CANDECOMP/PARAFAC (CP) decomposition. Drineas and Mahoney [11], Zhou and Cichocki [32], Da Costa et al. [9] and Tsourakakis [30] propose different randomized methods for computing HOSVD. The method in [30], which is called MACH, is also extended to computing HOOI. Mahoney et al. [23] and Caiafa and Cichocki [6] present results

that extend the CUR factorization for matrices to tensors. Other decomposition methods that only consider a small number of the tensor entries include those by Oseledets et al. [25] and Friedland et al. [13].

Another approach to decomposing large tensors is to use memory efficient and distributed methods. Kolda and Sun [19] introduce the Memory Efficient Tucker (MET) decomposition for sparse tensors as a solution to the so called intermediate blow-up problem which occurs when computing the chain of TTM products in HOOI. Other papers that use memory efficient and distributed methods include [4, 20, 21, 22, 28, 15, 1, 16, 24].

Other research focuses on handling streamed tensor data. Sun et al. [29] introduce a framework for incremental tensor analysis. The basic idea of their method is to find one set of factor matrices which works well for decomposing a sequence of tensors that arrive over time. Fanaee-T and Gama [12] introduce multi-aspect-streaming tensor analysis which is based on the histogram approximation concept rather than linear algebra techniques. Neither of these methods correspond to Tucker decomposition of a tensor whose elements are streamed. Gujral et al. [14] present a method for incremental CP decomposition.

We compare our algorithms to TUCKER-ALS and MET in Tensor Toolbox version 2.6 [3, 19], FSTD1 with adaptive index selection from [6], as well as the HOOI version of the MACH algorithm in [30].[1] TUCKER-ALS and MET, which are mathematically equivalent, provide good accuracy, but run out of memory as the tensor size increases. MACH scales somewhat better, but also runs out of memory for larger tensors. Its accuracy is also lower than that of TUCKER-ALS/MET. None of these algorithms are one-pass. FSTD1 scales well, but has accuracy issues on very sparse tensors. FSTD1 does not need to access all elements of the tensor and is one-pass, but since the entire tensor needs to be accessible the method cannot handle streamed data.

## 3 Tucker decomposition using TensorSketch

We now present our proposed algorithms. More detailed versions of them can be found in Section S3 of the supplement. A Matlab implementation of our algorithms can be found at https://github.com/OsmanMalik/tucker-tensorsketch.

### 3.1 First proposed algorithm: TUCKER-TS

For our first algorithm, we TENSORSKETCH both the least-squares problems in (3) and (4), and then solve the smaller resulting problems. We give an algorithm for this approach in Algorithm 2. We call it TUCKER-TS, where "TS" stands for TENSORSKETCH. The core tensor and factor matrices in line 1 are initialized randomly with each element i.i.d. Uniform$(-1, 1)$. The factor matrices are subsequently orthogonalized. On line 2 we define TENSORSKETCH operators of appropriate size. This is done by first defining COUNTSKETCH operators $\mathbf{S}_1^{(n)} \in \mathbb{R}^{J_1 \times I_n}$ and $\mathbf{S}_2^{(n)} \in \mathbb{R}^{J_2 \times I_n}$ for $n \in [N]$, as explained in Section 1.2. Then each operator $\mathbf{T}^{(n)}$, for $n \in [N]$, is defined as in (5) but based on $\{\mathbf{S}_1^{(n)}\}_{n \in [N]}$ and with the $n$th term excluded in the Kronecker and Khatri-Rao products. $\mathbf{T}^{(N+1)}$ is defined similarly, but based on $\{\mathbf{S}_2^{(n)}\}_{n \in [N]}$ and without excluding any terms in the Kronecker and Khatri-Rao products. The reason we use two different sets $\{\mathbf{S}_1^{(n)}\}_{n \in [N]}$ and $\{\mathbf{S}_2^{(n)}\}_{n \in [N]}$ of COUNTSKETCH operators with different target sketch dimensions $J_1$ and $J_2$, respectively, is that the design matrix in (4) has more rows than that in (3). In practice, this means that we choose $J_2 > J_1$. In Section 4 we provide some guidance on how to choose $J_1$ and $J_2$. We also want to point out that none of the sketch operators are stored explicitly as matrices in our implementation. Instead, we only generate and store the function $h$ and the diagonal of $\mathbf{D}$, which were defined in Section 1.2, for each COUNTSKETCH operator. We then use the formula in (5) when applying one of the TENSORSKETCH operators to a Kronecker product matrix. The computations

$\mathbf{T}^{(n)}\mathbf{Y}_{(n)}^{\top}$ and $\mathbf{T}^{(N+1)}\mathbf{y}_{(:)}$ on lines 5 and 7 cannot be done using the formula in (5), but are still computed implicitly without forming any full sketching matrices.

Since all sketch operators used on line 5 are defined in terms of the same set $\{\mathbf{S}_1^{(n)}\}_{n\in[N]}$, the least-squares problem on all iterations of that line except the first will depend in some way on the sketch operator $\mathbf{T}^{(n)}$ being applied. A similar dependence will exist between the least-squares problem on line 7 and $\mathbf{T}^{(N+1)}$ beyond the first iteration. It is important to note that the guarantees for TENSORSKETCHED least-squares in [10] hold when the random sketch is independent of the least-squares problem it is applied to. For these guarantees to hold, we would need to define a new TENSORSKETCH operator each time a least-squares problem is solved in Algorithm 2. In all of our experiments, we observe that our approach of instead defining the sketch operators upfront leads to a substantial reduction in the error for the algorithm as a whole (see Figure 1). We have not yet been able to provide theoretical justification for why this is.

The following proposition shows that the normal equations formulation of the least-squares problem on line 7 in Algorithm 2 is well-conditioned with high probability if $J_2$ is sufficiently large, and therefore can be efficiently solved using conjugate gradient (CG). This is true because the factor matrices are orthogonal, and does not hold for the smaller system on line 5, so this system we solve via direct methods. In our experiments, for an accuracy of 1e-6, CG takes about 15 iterations regardless of $I$. A proof of Proposition 3.1 is provided in Section S4 of the supplementary material.

**Proposition 3.1.** *Assume* $\mathbf{T}^{(N+1)}$ *is defined as in line 2 in Algorithm 2. Let* $\mathbf{M} \overset{\text{def}}{=} (\mathbf{T}^{(N+1)} \bigotimes_{i=N}^{1} \mathbf{A}^{(i)})^{\top} (\mathbf{T}^{(N+1)} \bigotimes_{i=N}^{1} \mathbf{A}^{(i)})$, *where all* $\mathbf{A}^{(n)}$ *have orthonormal columns, and suppose* $\varepsilon, \delta \in (0,1)$. *If* $J_2 \geq (\prod_n R_n)^2(2 + 3^N)/(\varepsilon^2 \delta)$, *then the 2-norm condition number of* $\mathbf{M}$ *satisfies* $\kappa(\mathbf{M}) \leq (1 + \varepsilon)^2/(1 - \varepsilon)^2$ *with at least probability* $1 - \delta$.

**Remark 3.2.** Defining the sketching operators upfront allows us to make the following improvements:

(a) Since $\mathcal{Y}$ remains unchanged throughout the algorithm, the $N + 1$ sketches of $\mathcal{Y}$ only need to be computed once, which we do upfront in a single pass over the data (using a careful implementation). This can also be done if elements of $\mathcal{Y}$ are streamed.

(b) Since the same COUNTSKETCH is applied to each $\mathbf{A}^{(n)}$ when sketching the Kronecker product in the inner loop, we can compute the quantity $\hat{\mathbf{A}}_{s_1}^{(n)} \overset{\text{def}}{=} (\text{FFT}(\mathbf{S}_1^{(n)}\mathbf{A}^{(n)}))^{\top}$ after updating $\mathbf{A}^{(n)}$ and reuse it when computing other factor matrices until $\mathbf{A}^{(n)}$ is updated again.

(c) When $I_n \geq J_1 + J_2$ for some $n \in [N]$, we can reduce the size of the least-squares problem on line 5. Note that the full matrix $\mathbf{A}^{(n)}$ is not needed until the return statement—only the sketches $\mathbf{S}_1^{(n)}\mathbf{A}^{(n)}$ and $\mathbf{S}_2^{(n)}\mathbf{A}^{(n)}$ are necessary to compute the different TENSORSKETCHES. Replacing $\mathbf{T}^{(n)}\mathbf{Y}_{(n)}^{\top}$ on line 5 with $\mathbf{T}^{(n)}[\mathbf{Y}_{(n)}^{\top}\mathbf{S}_1^{(n)\top}, \ \mathbf{Y}_{(n)}^{\top}\mathbf{S}_2^{(n)\top}]$, which also can be computed upfront, we get a smaller least-squares problem which has the solution $[\mathbf{S}_1^{(n)}\mathbf{A}^{(n)}, \ \mathbf{S}_2^{(n)}\mathbf{A}^{(n)}]$. Before the return statement, we then compute the full factor matrix $\mathbf{A}^{(n)}$. With this adjustment, we cannot orthogonalize the factor matrices on each iteration, and therefore Proposition 3.1 does not apply. In this case, we therefore use a dense method instead of CG when computing $\mathcal{G}$ in Algorithm 2.

### 3.2 Second proposed algorithm: TUCKER-TTMTS

We can rewrite the TTM product on line 4 of Algorithm 1 to $\mathbf{Z}_{(n)} = \mathbf{Y}_{(n)} \bigotimes_{\substack{i=N \\ i \neq n}}^{1} \mathbf{A}^{(i)}$. We TENSORSKETCH this formulation as follows: $\tilde{\mathbf{Z}}_{(n)} = (\mathbf{T}^{(n)}\mathbf{Y}_{(n)}^{\top})^{\top}\mathbf{T}^{(n)} \bigotimes_{\substack{i=N \\ i \neq n}}^{1} \mathbf{A}^{(i)}$, $n \in [N]$, where each $\mathbf{T}^{(n)} \in \mathbb{R}^{J_1 \times \prod_{i \neq n} I_i}$ is a TENSORSKETCH operator with target dimension $J_1$. We can similarly sketch the computation on line 8 in Algorithm 1 using a TENSORSKETCH operator $\mathbf{T}^{(N+1)} \in \mathbb{R}^{J_2 \times \prod_i I_i}$ with target dimension $J_2$. Replacing the computations on lines 4 and 8 in Algorithm 1 with these sketched computations, we get our second algorithm which we call TUCKER-TTMTS, where "TTMTS" stands for "TTM TENSORSKETCH." The algorithm is given in Algorithm 3. The initialization of the factor matrices on line 1a and the definition of the sketching operators on line 1b are done in the same way as in Algorithm 2. Since the sketch operators are defined upfront here as well, the same caveat applies here as for Algorithm 2. The main benefit of TUCKER-TTMTS over TUCKER-TS is that it scales better with the target rank (see Section 3.4).

---
**Algorithm 2:** TUCKER-TS (proposal)
---

**input** : $\mathcal{Y}$, target rank $(R_1, R_2, \ldots, R_N)$, sketch dimensions $(J_1, J_2)$
**output** : Rank-$(R_1, R_2, \ldots, R_N)$ Tucker decomposition $[\![\mathcal{G}; \mathbf{A}^{(1)}, \ldots, \mathbf{A}^{(N)}]\!]$ of $\mathcal{Y}$

1 Initialize $\mathcal{G}, \mathbf{A}^{(2)}, \mathbf{A}^{(3)}, \ldots, \mathbf{A}^{(N)}$
2 Define TENSORSKETCH operators $\mathbf{T}^{(n)} \in \mathbb{R}^{J_1 \times \prod_{i \neq n} I_i}$, for $n \in [N]$, and $\mathbf{T}^{(N+1)} \in \mathbb{R}^{J_2 \times \prod_i I_i}$
3 **repeat**
4      **for** $n = 1, \ldots, N$ **do**
5          $\mathbf{A}^{(n)} = \arg\min_{\mathbf{A}} \left\| \left( \mathbf{T}^{(n)} \bigotimes_{i=N, i \neq n}^1 \mathbf{A}^{(i)} \right) \mathbf{G}_{(n)}^\top \mathbf{A}^\top - \mathbf{T}^{(n)} \mathbf{Y}_{(n)}^\top \right\|_F^2$
6      **end**
7      $\mathcal{G} = \arg\min_{\mathbf{z}} \left\| \left( \mathbf{T}^{(N+1)} \bigotimes_{i=N}^1 \mathbf{A}^{(i)} \right) \mathbf{z}_{(:)} - \mathbf{T}^{(N+1)} \mathbf{y}_{(:)} \right\|_2^2$
8      Orthogonalize each $\mathbf{A}^{(i)}$ and update $\mathcal{G}$
9 **until** *termination criteria met*
10 **return** $\mathcal{G}, \mathbf{A}^{(1)}, \ldots, \mathbf{A}^{(N)}$

---

The following informal proposition shows that the error for each sketched computation in TUCKER-TTMTS is additive rather than multiplicative as for TUCKER-TS. A formal statement and proof are given in Section S5 of the supplementary material.

**Proposition 3.3** (TUCKER-TTMTS (informal)). *Assume each* TENSORSKETCH *operator is redefined prior to being used. Let* OBJ *denote the objective function in* (3), *and let* $\tilde{\mathbf{A}}^{(n)}$ *be the* $R_n$ *leading left singular vectors of* $\mathbf{Z}_{(n)}$ *defined on line 4 in Algorithm 3. Under certain conditions,* $\tilde{\mathbf{A}}^{(n)}$ *satisfies* $\text{OBJ}(\tilde{\mathbf{A}}^{(n)}) \leq \min_{\mathbf{A}} \text{OBJ}(\mathbf{A}) + \varepsilon C$ *with high probability if* $J_1$ *is sufficiently large, where* $C$ *depends on* $\mathcal{Y}$, *the target rank, and the other factor matrices. A similar result holds for the update on line 8 and the objective function in* (4).

---
**Algorithm 3:** TUCKER-TTMTS (proposal)
---

```
/* Identical to Algorithm 1, except for the lines below */
```

1a Initialize $\mathbf{A}^{(2)}, \mathbf{A}^{(3)}, \ldots, \mathbf{A}^{(N)}$
1b Define TENSORSKETCH operators $\mathbf{T}^{(n)} \in \mathbb{R}^{J_1 \times \prod_{i \neq n} I_i}$, for $n \in [N]$, and $\mathbf{T}^{(N+1)} \in \mathbb{R}^{J_2 \times \prod_i I_i}$

4 $\mathbf{Z}_{(n)} = \left( \mathbf{T}^{(n)} \mathbf{Y}_{(n)}^\top \right)^\top \left( \mathbf{T}^{(n)} \bigotimes_{i=N, i \neq n}^1 \mathbf{A}^{(i)} \right)$

8 $\mathbf{g}_{(:)} = \left( \mathbf{T}^{(N+1)} \bigotimes_{i=N}^1 \mathbf{A}^{(i)} \right)^\top \mathbf{T}^{(N+1)} \mathbf{y}_{(:)}$

---

### 3.3 Stopping conditions and orthogonalization

Unless stated otherwise, we stop after 50 iterations or when the change in $\|\mathcal{G}\|$ is less than 1e-3. The same type of convergence criteria are used in [19]. In Algorithm 2, we orthogonalize the factor matrices and update $\mathcal{G}$ using the reduced QR factorization. If we use the improvement in Remark 3.2 (c), we need to approximate $\mathcal{G}$. This is discussed in Section S3.1 of the supplementary material. In Algorithm 3, we compute an estimate of $\mathcal{G}$ using the same formula as in line 8, but using the smaller sketch dimension $J_1$ instead. Unlike in TUCKER-ALS, the objective is not guaranteed to decrease on each iteration of our algorithms. Despite this, the only practical different between our algorithms and TUCKER-ALS is that the tolerance may need to be set differently.

We would like to point out that we cannot provide global convergence guarantees for our algorithms. Although a global analysis would be desirable, it is important to note that such an analysis is difficult even for TUCKER-ALS. Indeed, TUCKER-ALS is not guaranteed to converge to the global optimum or even a stationary point (see Section 4.2 in [18]).

## 3.4 Complexity analysis

We compare the complexity of Algorithms 1–3, FSTD1 and MACH for the case $N = 3$. We assume that $I_n = I$ and $R_n = R < I$ for all $n \in [N]$. Furthermore, we assume that $J_1 = KR^{N-1}$ and $J_2 = KR^N$ for some constant $K > 1$, which is a choice that works well in practice. Table 1 shows the complexity when each of the variables $I$ and $R$ are assumed to be large. A more detailed complexity analysis of the proposed algorithms is given in Section S3.2 of the supplementary material.

|  | Variable assumed to be large | |
| Algorithm | $I$ = size of fiber | $R$ = rank |
| --- | --- | --- |
| T.-ALS (Alg. 1) | $(\#iter + 1) \cdot RI^3$ | $(\#iter + 1) \cdot RI^3$ |
| FSTD1 [6] | $IR^4$ | $R^5$ |
| MACH [30] | $(\#iter + 1) \cdot RI^3$ | $(\#iter + 1) \cdot RI^3$ |
| T.-TS (proposal, Alg. 2) | $\text{nnz}(\mathcal{Y}) + IR^4$ | $R^3 + \#iter \cdot R^6$ |
| T.-TTMTS (proposal, Alg. 3) | $\text{nnz}(\mathcal{Y}) + IR^4 + \#iter \cdot IR^4$ | $R^6 + \#iter \cdot R^4$ |

Table 1: Leading order computational complexity, ignoring $\log$ factors and assuming $K = \mathcal{O}(1)$, where $\#iter$ is the number of main loop iterations. $\mathcal{Y}$ is the 3-way data tensor we decompose. The main benefits of our proposed algorithms is reducing the $\mathcal{O}(I^N)$ complexity of Algorithm 1 to $R^{\mathcal{O}(N)}$ complexity due to the sketching, since typically $R \ll I$. The complexity of MACH is the same as that of TUCKER-ALS, but with a smaller constant factor.

# 4 Experiments

In this section we present results from experiments. Our Matlab implementation that we provided a link to at the beginning of Section 3 comes with demo script files for running experiments similar to those presented here. All synthetic results are averages over ten runs in an environment using four cores of an Intel Xeon E5-2680 v3 @2.50GHz CPU and 21 GB of RAM. For Algorithms 2 and 3, the sketch dimensions $J_1$ and $J_2$ must be chosen. We have found that the choice $J_1 = KR^{N-1}$ and $J_2 = KR^N$, for a constant $K > 4$, works well in practice. Figure 1 shows examples of how the error of TUCKER-TS and TUCKER-TTMTS, in relation to that of TUCKER-ALS, changes with $K$. It also shows results for variants of each algorithm for which the TENSORSKETCH operator is redefined each time it is used (called "multi-pass" in the figure). For both algorithms, defining TENSORSKETCH operators upfront leads to higher accuracy than redefining them before each application. In subsequent experiments, we always define the sketch operators upfront (i.e., as written in Algorithms 2 and 3) and, unless stated otherwise, always use $K = 10$.

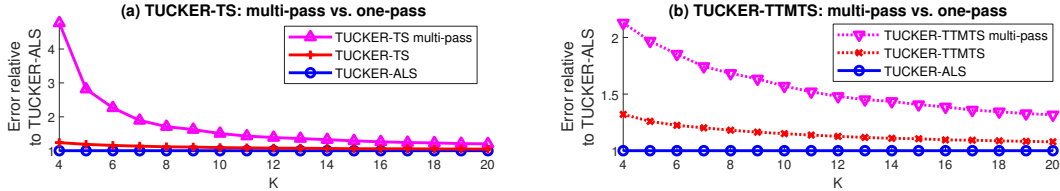

Figure 1: Errors of TUCKER-TS and TUCKER-TTMTS, relative to that of TUCKER-ALS, for different values of the sketch dimension parameter $K$. For both plots, the tensor size is $500 \times 500 \times 500$ with $\text{nnz}(\mathcal{Y}) \approx$ 1e+6 and true rank $(15, 15, 15)$. The algorithms use a target rank of $(10, 10, 10)$.

## 4.1 Sparse synthetic data

In this subsection, we apply our algorithms to synthetic sparse tensors. For all synthetic data we use $I_n = I$ and $R_n = R$ for all $n \in [N]$. The sparse tensors are each created from a random dense core tensor and random sparse factor matrices, where the sparsity of the factor matrices is chosen

to achieve the desired sparsity of the tensor. We add i.i.d. normally distributed noise with standard deviation 1e-3 to all nonzero tensor elements.

Figures 2 and 3 show how the algorithms scale with *increased dimension* size $I$. Figure 4 and 5 show how the algorithms scale with tensor *density* and algorithm *target rank* $R$, respectively. TUCKER-ALS/MET and MACH run out of memory when $I = $ 1e+5. FSTD1 is fast and scalable but inaccurate for very sparse tensors. The algorithm repeatedly finds indices of $\mathcal{Y}$ by identifying the element of maximum magnitude in fibers of the residual tensor. However, when $\mathcal{Y}$ is very sparse, it frequently happens that whole fibers in the residual tensor are zero. In those cases, the algorithm fails to find a good set of indices. This explains its poor accuracy in our experiments. We see that TUCKER-TS performs very well when $\mathcal{Y}$ truly is low-rank and we use that same rank for reconstruction. TUCKER-TTMTS in general has a larger error than TUCKER-TS, but scales better with higher target rank. Moreover, when the true rank of the input tensor is greater than the target rank (Figure 3), which is closer to what real data might look like, the error of TUCKER-TTMTS is much closer to that of TUCKER-TS.

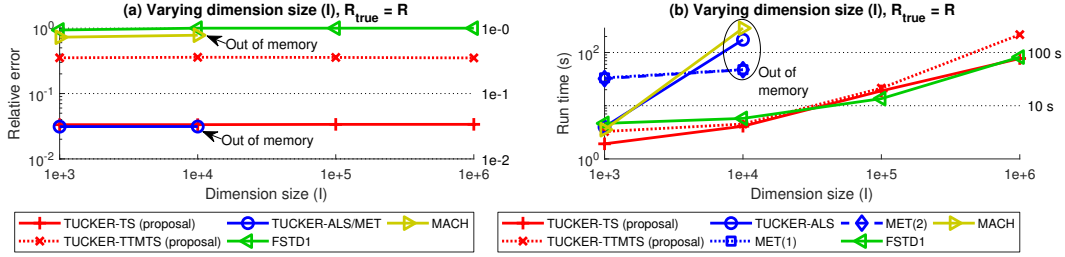

Figure 2: Relative error and run time for random sparse 3-way tensors with varying dimension size $I$ and nnz($\mathcal{Y}$) $\approx$ 1e+6. Both the true and target ranks are $(10, 10, 10)$.

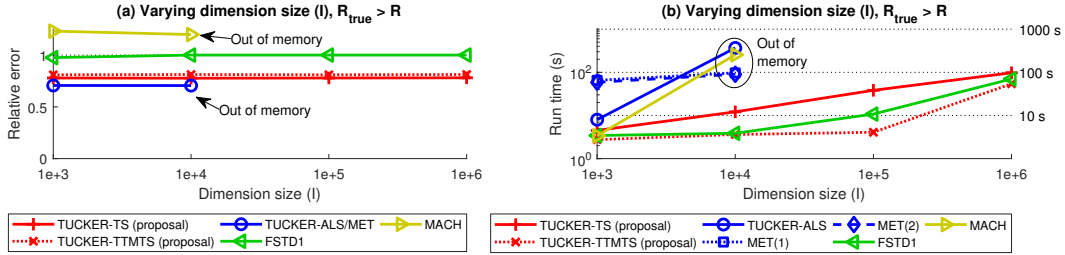

Figure 3: Relative error and run time for random sparse 3-way tensors with varying dimension size $I$ and nnz($\mathcal{Y}$) $\approx$ 1e+6. The true rank is $(15, 15, 15)$ and target rank is $(10, 10, 10)$. A convergence tolerance of 1e-1 is used for these experiments.

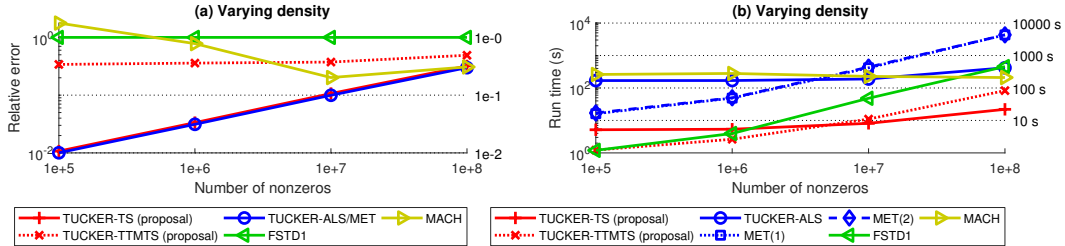

Figure 4: Relative error and run time for random sparse 3-way tensors with dimension size $I = $ 1e+4 and varying number of nonzeros. Both the true and target ranks are $(10, 10, 10)$.

## 4.2 Dense real-world data

In this section we apply TUCKER-TTMTS to a real dense tensor representing a grayscale video. The video consists of 2,200 frames, each of size 1,080 by 1,980 pixels. The whole tensor, which

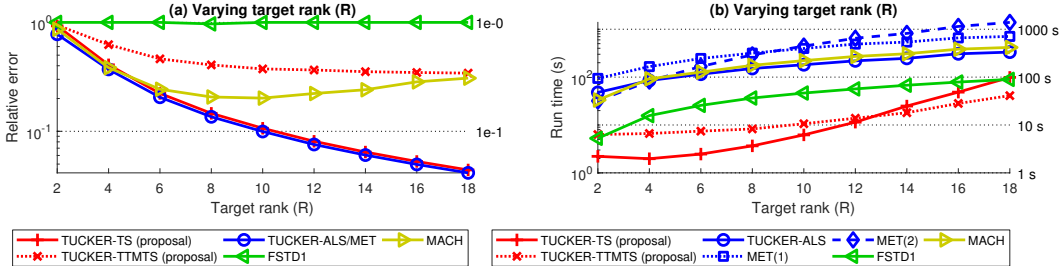

Figure 5: Relative error and run time for random sparse 3-way tensors with dimension size $I = 1e+4$ and $\text{nnz}(\mathcal{Y}) \approx 1e+7$. The true and target ranks are $(R, R, R)$, with $R$ varying.

requires 38 GB of RAM, is too large to load at the same time. Instead, it is loaded in pieces which are sketched and then added together. The video shows a natural scene, which is disturbed by a person passing by the camera twice. Since the camera is in a fixed position, we can expect this tensor to be compressible. We compute a rank $(10, 10, 10)$ Tucker decomposition of the tensor using TUCKER-TTMTS with the sketch dimension parameter set to $K = 100$ and a maximum of 30 iterations. We then apply k-means clustering to the factor matrix $\mathbf{A}^{(3)} \in \mathbb{R}^{2200 \times 10}$ corresponding to the time dimension, classifying each frame using the corresponding row in $\mathbf{A}^{(3)}$ as a feature vector. We find that using three clusters works better than using two. We believe this is due to the fact that the light intensity changes through the video due to clouds, which introduces a third time varying factor. Figure 6 shows five sample frames with the corresponding assigned clusters. With few exceptions, the frames which contain a disturbance are correctly grouped together into class 3 with the remaining frames grouped into classes 1 and 2. The video experiment is online and a link to it is provided at https://github.com/OsmanMalik/tucker-tensorsketch.

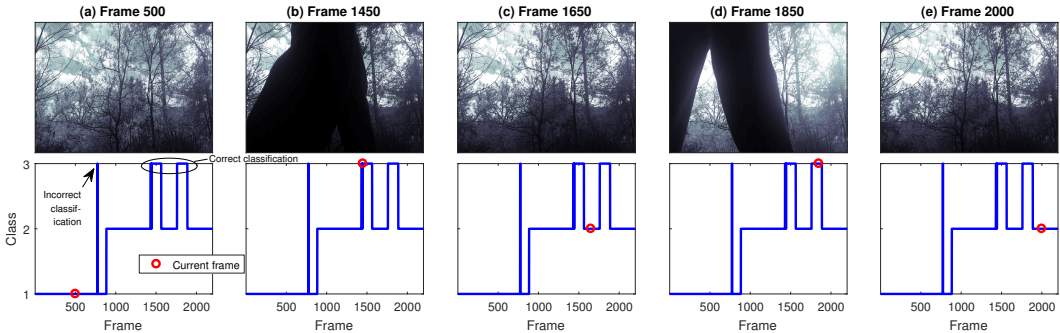

Figure 6: Five sample frames with their assigned classes. The frames (b) and (d) contain a disturbance.

## 5   Conclusion

We have proposed two algorithms for low-rank Tucker decomposition which incorporate TENSORS-KETCH and can handle streamed data. Experiments corroborate our complexity analysis which shows that the algorithms scale well both with dimension size and density. TUCKER-TS, and to a lesser extent TUCKER-TTMTS, scale poorly with target rank, so they are most useful when $R \ll I$.

## Acknowledgments

We would like to thank the reviewers for their many helpful comments and suggestions which helped improve this paper.

This material is based upon work supported by the National Science Foundation under Grant No. 1810314.

This work utilized the RMACC Summit supercomputer, which is supported by the National Science Foundation (awards ACI-1532235 and ACI-1532236), the University of Colorado Boulder, and Colorado State University. The Summit supercomputer is a joint effort of the University of Colorado Boulder and Colorado State University.

## Footnotes

[1]For FSTD1, we use the Matlab code from the website of one of the authors (http://ccaiafa.wixsite.com/cesar). For MACH, we adapted the Python code provided on the author's website (https://tsourakakis.com/mining-tensors/) to Matlab. MACH requires an algorithm for computing the HOOI decomposition of the sparsified tensor. For this, we use TUCKER-ALS and then switch to higher orders of MET as necessary when we run out of memory. As recommended in [30], we keep each nonzero entry in the original tensor with probability 0.1 when using MACH.

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
