[Supplementary Material]

# Supplementary Material

# Low-Rank Tucker Decomposition of Large Tensors Using TensorSketch

**Osman Asif Malik**
Department of Applied Mathematics
University of Colorado Boulder
`osman.malik@colorado.edu`

**Stephen Becker**
Department of Applied Mathematics
University of Colorado Boulder
`stephen.becker@colorado.edu`

## S1 Definitions of tensor concepts

In this section we give definitions of the tensor concepts used in the main manuscript and in the supplementary material. We use the same notations and definitions as in [7]. A tensor $\mathcal{X} \in \mathbb{R}^{I_1 \times I_2 \times \cdots \times I_N}$ is an array of dimension $N$, also called an $N$-way array. Boldface Euler script letters, e.g. $\mathcal{X}$, will denote tensors of dimension 3 or greater; bold capital letters, e.g. $\mathbf{X}$, will denote matrices; bold lowercase letters, e.g. $\mathbf{x}$, will denote vectors; and lowercase letters, e.g. $x$, will denote scalars. We use a colon to denote all elements along a certain dimension; for example, $\mathbf{x}_{n:}$ is the $n$th row of $\mathbf{X}$, $\mathbf{x}_{:n}$ is the $n$th column of $\mathbf{X}$, and $\mathbf{X}_{::n}$ is the $n$th so called frontal slice of the 3-way tensor $\mathcal{X}$. The *norm* of a tensor $\mathcal{X} \in \mathbb{R}^{I_1 \times I_2 \times \cdots \times I_N}$ is defined as

$$\|\mathcal{X}\| = \sqrt{\sum_{i_1=1}^{I_1} \sum_{i_2=1}^{I_2} \cdots \sum_{i_N=1}^{I_N} x_{i_1 i_2 \cdots i_N}^2}. \tag{S1}$$

The *Kronecker product* of two matrices $\mathbf{A} \in \mathbb{R}^{I_1 \times R_1}$ and $\mathbf{B} \in \mathbb{R}^{I_2 \times R_2}$ is denoted by $\mathbf{A} \otimes \mathbf{B} \in \mathbb{R}^{I_1 I_2 \times R_1 R_2}$ and is defined by

$$\mathbf{A} \otimes \mathbf{B} = \begin{bmatrix} a_{11}\mathbf{B} & a_{12}\mathbf{B} & \cdots & a_{1R_1}\mathbf{B} \\ a_{21}\mathbf{B} & a_{22}\mathbf{B} & \cdots & a_{2R_1}\mathbf{B} \\ \vdots & \vdots & & \vdots \\ a_{I_1 1}\mathbf{B} & a_{I_1 2}\mathbf{B} & \cdots & a_{I_1 R_1}\mathbf{B} \end{bmatrix}. \tag{S2}$$

The *Khatri-Rao product* of two matrices $\mathbf{A} \in \mathbb{R}^{I_1 \times R}$ and $\mathbf{B} \in \mathbb{R}^{I_2 \times R}$ is denoted by $\mathbf{A} \odot \mathbf{B} \in \mathbb{R}^{I_1 I_2 \times R}$ and is defined by

$$\mathbf{A} \odot \mathbf{B} = \begin{bmatrix} \mathbf{a}_{:1} \otimes \mathbf{b}_{:1} & \mathbf{a}_{:2} \otimes \mathbf{b}_{:2} & \cdots & \mathbf{a}_{:R} \otimes \mathbf{b}_{:R} \end{bmatrix}. \tag{S3}$$

We can "flatten," or matricize, a tensor into a matrix. The *mode-n matricization* of a tensor $\mathcal{X} \in \mathbb{R}^{I_1 \times I_2 \times \cdots \times I_N}$ is denoted by $\mathbf{X}_{(n)} \in \mathbb{R}^{I_n \times \prod_{i \neq n} I_i}$ and maps the element on position $(i_1, i_2, \ldots, i_N)$ in $\mathcal{X}$ to position $(i_n, j)$ in $\mathbf{X}_{(n)}$, where

$$j = 1 + \sum_{\substack{k=1 \\ k \neq n}}^{N} (i_k - 1)J_k \quad \text{with} \quad J_k = \prod_{\substack{m=1 \\ m \neq n}}^{k-1} I_m. \tag{S4}$$

Similarly, $\mathbf{x}_{(:)} \in \mathbb{R}^{\prod_n I_n}$ will denote the vectorization of $\mathcal{X}$ we get by stacking the columns of $\mathbf{X}_{(1)}$ into a long column vector. The *n-mode product* of a tensor $\mathcal{X} \in \mathbb{R}^{I_1 \times I_2 \times \cdots \times I_N}$ and matrix

$\mathbf{A} \in \mathbb{R}^{J \times I_n}$ is denoted by $\mathcal{X} \times_n \mathbf{A} \in \mathbb{R}^{I_1 \times \cdots \times I_{n-1} \times J \times I_{n+1} \times \cdots \times I_N}$ and defined by

$$(\mathcal{X} \times_n \mathbf{A})_{i_1 \cdots i_{n-1} j i_{n+1} \cdots i_N} = \sum_{i_n=1}^{I_n} x_{i_1 i_2 \cdots i_N} a_{j i_n}. \tag{S5}$$

We can express this definition more compactly using matrix notation as follows:

$$\mathcal{Y} = \mathcal{X} \times_n \mathbf{A} \quad \Leftrightarrow \quad \mathbf{Y}_{(n)} = \mathbf{A}\mathbf{X}_{(n)}. \tag{S6}$$

## S2    Some further information on CountSketch and TensorSketch

As mentioned in the main manuscript, TENSORSKETCH can be seen as restricted kind of COUNTS-KETCH which can be applied very efficiently to Kronecker products. We first explain how COUNTS-KETCH can be applied to least-squares problems and for matrix multiplication, and then explain how these ideas extend to TENSORSKETCH. Nothing in this subsection is our own work; it is simply a brief introduction to the sketching techniques we use.

The basic idea of applying COUNTSKETCH to least-squares regression is the following. Suppose $\mathbf{A} \in \mathbb{R}^{I \times R}$, where $I \gg R$, and $\mathbf{y} \in \mathbb{R}^I$, and consider solving the overdetermined least-squares problem

$$\mathbf{x}^* \overset{\text{def}}{=} \underset{\mathbf{x} \in \mathbb{R}^R}{\arg\min} \|\mathbf{A}\mathbf{x} - \mathbf{y}\|_2. \tag{S7}$$

COUNTSKETCH allows us to reduce the size of this problem by first applying a subspace embedding matrix $\mathbf{S} : \mathbb{R}^I \to \mathbb{R}^J$ to $\mathbf{A}$ and $\mathbf{y}$, where $J$ is much smaller than $I$, and so that we instead solve

$$\mathbf{x}' \overset{\text{def}}{=} \underset{\mathbf{x} \in \mathbb{R}^R}{\arg\min} \|\mathbf{S}\mathbf{A}\mathbf{x} - \mathbf{S}\mathbf{y}\|_2. \tag{S8}$$

One way to define the matrix $\mathbf{S}$ is as $\mathbf{S} = \mathbf{P}\mathbf{D}$, where

- $\mathbf{P} \in \mathbb{R}^{J \times I}$ is a matrix with $p_{h(i),i} = 1$, and all other entries equal to 0;
- $h : [I] \to [J]$ is a random map such that $(\forall i \in [I])(\forall j \in [J]) \, \mathbb{P}(h(i) = j) = 1/J$; and
- $\mathbf{D} \in \mathbb{R}^{I \times I}$ is a diagonal matrix, with each diagonal entry equal to $+1$ or $-1$ with equal probability.

For a fixed $\varepsilon > 0$, one can ensure that the solution to the sketched problem (S8) satisfies

$$\|\mathbf{A}\mathbf{x}' - \mathbf{y}\|_2 \leq (1 + \varepsilon) \|\mathbf{A}\mathbf{x}^* - \mathbf{y}\|_2 \tag{S9}$$

with high probability by choosing the sketch dimension $J$ sufficiently large. Moreover, the matrix $\mathbf{S}$ can be applied in $\mathcal{O}(\text{nnz}(\mathbf{A}))$ time, where $\text{nnz}(\mathbf{A})$ denotes the number of nonzero elements of $\mathbf{A}$, making it an especially efficient sketch if $\mathbf{A}$ is sparse; see [2] for further details.

COUNTSKETCH can also be used for approximate matrix multiplication. Let $\mathbf{S} \in \mathbb{R}^{J \times I}$ denote the COUNTSKETCH operator. Informally, for matrices $\mathbf{A}$ and $\mathbf{B}$ with $I$ rows and a fixed $\varepsilon > 0$, one can ensure that

$$\left\| \mathbf{A}^\top \mathbf{S}^\top \mathbf{S} \mathbf{B} - \mathbf{A}^\top \mathbf{B} \right\|_F \leq \varepsilon \|\mathbf{A}\|_F \|\mathbf{B}\|_F \tag{S10}$$

with high probability by choosing the sketch dimension $J$ sufficiently large. A formal statement and proof of this is given in Lemma 32 of [2].

Now suppose the matrix $\mathbf{A}$ is of the form $\mathbf{A} = \mathbf{A}^{(1)} \otimes \mathbf{A}^{(2)} \otimes \cdots \otimes \mathbf{A}^{(N)}$, where each $\mathbf{A}^{(n)} \in \mathbb{R}^{I_n \times R_n}$, $I_n > R_n$. Then $\mathbf{A}$ is of size $I \times R$, where $I \overset{\text{def}}{=} \prod_n I_n$ and $R \overset{\text{def}}{=} \prod_n R_n$. TENSORSKETCH is a restricted variant of COUNTSKETCH which allows us to sketch $\mathbf{A}$ without having to first form the matrix. This is done by instead sketching each factor matrix $\mathbf{A}^{(n)}$ individually, and then computing the corresponding sketch of $\mathbf{A}$, which can be done efficiently using the fast Fourier transform (FFT).

TENSORSKETCH was first introduced in [8] where it is applied to compressed matrix multiplication. In [9], TENSORSKETCH is used for approximating support vector machine polynomial kernels efficiently. Avron et al. [1] show that TENSORSKETCH provides an oblivious subspace embedding. Diao et al. [3] show that an approximate solution to the least-squares problem in the sense of (S9) when $\mathbf{A}$ is a Kronecker product can be obtained with high probability by solving the TENSORSKETCHED

problem instead of the full problem. The remainder of this subsection will briefly describe how TENSORSKETCH works.

In the following, in order to adhere to MATLAB notation with indexing starting at 1, some definitions will differ slightly from those in [3].

**Definition S2.1** ($k$-wise independent)**.** A family $\mathcal{H} \stackrel{\text{def}}{=} \{h : [I] \to [J]\}$ of hash functions is $k$-*wise independent* if for any distinct $x_1, \ldots, x_k \in [I]$, and uniformly random $h \in \mathcal{H}$, the hash codes $h(x_1), \ldots, h(x_k)$ are independent random variables, and the hash code of any fixed $x \in [I]$ is uniformly distributed in $[J]$. We also call a function $h$ drawn randomly and uniformly from $\mathcal{H}$ $k$-wise independent.

For $n \in [N]$, let

- $h_n : [I_n] \to [J]$ be 3-wise independent hash functions, and let
- $s_n : [I_n] \to \{-1, +1\}$ be 4-wise independent sign functions.

Furthermore, define the hash function

$$H : [I_1] \times [I_2] \times \cdots \times [I_N] \to [J] : (i_1, \ldots, i_N) \mapsto \left( \sum_{n=1}^{N} (h_n(i_n) - 1) \mod J \right) + 1 \quad \text{(S11)}$$

and the sign function

$$S : [I_1] \times [I_2] \times \cdots \times [I_N] \to \{-1, 1\} : (i_1, \ldots, i_N) \mapsto \prod_{n=1}^{N} s_n(i_n). \quad \text{(S12)}$$

TENSORSKETCH is the same as applying COUNTSKETCH to $\mathbf{A}$ with $\mathbf{P}$ defined using $H$ instead of $h$, and with the diagonal of $\mathbf{D}$ given by $S$. To understand how the maps $H$ and $S$ map a certain row index, note that there is a bijection between the set of rows $i \in [\prod_n I_n]$ of $\mathbf{A}$ and the $N$-tuples $(i_1, i_2, \ldots, i_N) \in [I_1] \times [I_2] \times \cdots \times [I_N]$. Specifically, for the $i$th row, there is a unique $N$-tuple $(i_1, i_2, \ldots, i_N)$ such that $\mathbf{a}_{i:} = \mathbf{a}_{i_1:}^{(1)} \otimes \mathbf{a}_{i_2:}^{(2)} \otimes \cdots \otimes \mathbf{a}_{i_N:}^{(N)}$. Pagh [8] observed that the application of COUNTSKETCH based on $H$ and $S$ can be done efficiently—without ever forming $\mathbf{A}$—by COUNTSKETCHING each factor matrix $\mathbf{A}^{(n)}$ using $h_n$ and $s_n$, and then computing the TENSORSKETCH of $\mathbf{A}$ using FFT. Let $\mathbf{S}^{(n)}$ be the COUNTSKETCH matrix corresponding to using the hash function $h_n$ and the diagonal matrix $\mathbf{D}^{(n)}$ with diagonal $d_{ii}^{(n)} = s_n(i)$. The application of $\mathbf{S}^{(n)}$ to $\mathbf{a}_{:r_n}^{(n)}$, the $r_n$th column of $\mathbf{A}^{(n)}$, can be represented by the $J-1$ degree polynomial

$$\mathcal{P}_{n,r_n}(\omega) = \sum_{i=1}^{I_n} s_n(i) a_{ir_n}^{(n)} \omega^{h_n(i)-1} = \sum_{j=1}^{J} c_{jr_n}^{(n)} \omega^{j-1}, \quad \text{(S13)}$$

where $\mathbf{c}_{r_n}^{(n)} \stackrel{\text{def}}{=} (c_{1r_n}^{(n)}, \ldots, c_{Jr_n}^{(n)})$ are the polynomial coefficients. With this representation, the elements of $\mathbf{a}_{:r_n}^{(n)}$ are multiplied with the correct sign given by $s_n$, and then grouped together in different polynomial coefficients depending on which bucket $h_n$ assigns them to. Letting $\mathbf{T}$ denote the TENSORSKETCH operator, the $r$th column of $\mathbf{TA}$ can similarly be represented as the polynomial

$$\mathcal{P}_r(\omega) = \sum_{i=1}^{I} S(i_1, \ldots, i_N) a_{ir} \omega^{H(i_1,\ldots,i_N)} \quad \text{(S14)}$$

$$= \sum_{i=1}^{I} s_1(i_1) \cdots s_N(i_N) a_{i_1 r_1}^{(1)} \cdots a_{i_N r_N}^{(N)} \omega^{h_1(i_1)+\cdots+h_N(i_N)-N \mod J} \quad \text{(S15)}$$

$$= \text{FFT}^{-1} \left( \text{FFT} \left( \mathbf{c}_{r_1}^{(1)} \right) * \cdots * \text{FFT} \left( \mathbf{c}_{r_N}^{(N)} \right) \right), \quad \text{(S16)}$$

where FFT denotes the unpadded FFT, "$*$" denotes Hadamard (element-wise) product and where the $i$ in (S14) corresponds to the $N$-tuple $(i_1, i_2, \ldots, i_N) \in [I_1] \times [I_2] \times \cdots \times [I_N]$. Similarly, $r$ corresponds to the $N$-tuple $(r_1, r_2, \ldots, r_N) \in [R_1] \times [R_2] \times \cdots \times [R_N]$. So each column of $\mathbf{TA}$

can be computed efficiently from the sketches of the corresponding columns of the factor matrices. More concisely, we can write

$$\mathbf{TA} = \mathrm{FFT}^{-1}\left(\left(\bigodot_{n=1}^{N}\left(\mathrm{FFT}\left(\mathbf{S}^{(n)}\mathbf{A}^{(n)}\right)\right)^{\top}\right)^{\top}\right), \tag{S17}$$

where FFT is applied columnwise to a matrix argument. Applying $\mathbf{T}$ naively to $\mathbf{A}$ would, in the general dense case, cost $\mathcal{O}\left(J\prod_{n=1}^{N}I_n R_n\right)$. With the trick above, however, it is straightforward to compute the reduced cost to be $\mathcal{O}\left(J\sum_{n=1}^{N}I_n R_n + J\log J\sum_{n=1}^{N}R_n + J\log J\prod_{n=1}^{N}R_n\right)$.

Theorem 3.1 in [3] gives theoretical guarantees for the optimality of the solution to (S8) when $\mathbf{S}$ is TENSORSKETCH. We have found the sketch dimension of that theorem to be pessimistic in practice, with much smaller sketch dimensions yielding satisfying results. Indeed, Example 6.1 of [3] also achieves good accuracy with a significantly smaller sketch dimension than what the theorem dictates.

Approximate matrix multiplication as in (S10) also holds for TENSORSKETCH; see Lemma B.1 in [3] for a formal statement and proof.

## S3   Detailed algorithms

In the detailed algorithms below, we will use the following notations: $\mathrm{CS}(\mathbf{A}, h, s)$ will denote the application of COUNTSKETCH to the matrix $\mathbf{A}$ using the hash function $h$ and the sign function $s$. $\mathrm{TS}\left(\mathbf{A}, \{h_n\}_{n=1}^{N}, \{s_n\}_{n=1}^{N}\right)$ will denote the application of TENSORSKETCH to the matrix $\mathbf{A}$ with the set of hash functions $\{h_n\}_{n=1}^{N}$ and sign functions $\{s_n\}_{n=1}^{N}$. We will overload the function TS in the following way. With the same notation as in (S17), suppose $\hat{\mathbf{A}}_s^{(n)} = \left(\mathrm{FFT}\left(\mathbf{S}^{(n)}\mathbf{A}^{(n)}\right)\right)^{\top}$. We will define TS with a single argument as

$$\mathrm{TS}\left(\left\{\hat{\mathbf{A}}_s^{(n)}\right\}_{n=1}^{N}\right) = \mathrm{FFT}^{-1}\left(\left(\bigodot_{n=1}^{N}\hat{\mathbf{A}}_s^{(n)}\right)^{\top}\right). \tag{S18}$$

In Algorithm S1, we give a version of TUCKER-TS where we define a new TENSORSKETCH operator before each least-squares sketch. In Algorithm S2, we give a detailed version of the TUCKER-TS algorithm in the main manuscript in the case when the dimension is sufficiently small for double-sketching to be unnecessary, i.e., when each $I_n \leq J_1 + J_2$ and the improvement for TUCKER-TS in Remark 3.2 (c) in the main manuscript is not used. In Algorithm S3, we give a detailed version of TUCKER-TS for when $I_n \geq J_1 + J_2$ for all $n \in [N]$ and the improvement in the remark is applied to all dimensions. In cases when some $I_n$ are large and some small, it is straightforward to combine elements from both algorithms to get the optimal algorithm. We give the algorithm for TUCKER-TS in these two versions to simplify complexity analysis. In Algorithm S4, we give a detailed version of the TUCKER-TTMTS algorithm in the main manuscript.

### S3.1   Orthogonalization

In Algorithms S1 and S2 we orthogonalize using QR factorization in the following way. The computation

$$\mathbf{Q}^{(n)}\mathbf{R}^{(n)} = \mathbf{A}^{(n)} \quad \text{reduced QR factorization} \tag{S19}$$

$$\mathbf{A}^{(n)} = \mathbf{Q}^{(n)} \tag{S20}$$

$$\mathcal{G} = \mathcal{G} \times_n \mathbf{R}^{(n)} \tag{S21}$$

ensures that each $\mathbf{A}^{(n)}$ is orthogonal, and that the Tucker decomposition remains unchanged. In Algorithm S3, we cannot do this computation since we only maintain sketched versions of the factor matrices. We will use the following heuristic instead. After updating $\mathcal{G}$ on line 13, we compute an approximation of the normalization of $\mathcal{G}$ as follows: Set $\mathcal{G}_{\text{normalized}} = \mathcal{G}$. For $n = 1, \ldots, N$,

$$\mathbf{Q}^{(n)}\mathbf{R}^{(n)} = \mathbf{A}_{s_2}^{(n)} \quad \text{reduced QR factorization} \tag{S22}$$

$$\mathcal{G}_{\text{normalized}} = \mathcal{G}_{\text{normalized}} \times_n \mathbf{R}^{(n)}. \tag{S23}$$

We then check for convergence based on $\|\mathcal{G}_{\text{normalized}}\|$ instead of $\|\mathcal{G}\|$.

## S3.2 Complexity analysis

We analyze the complexity of TUCKER-ALS given in the main paper, as well as Algorithms S1–S4. One can show that the leading order cost of applying TENSORSKETCH $\mathbf{T} \in \mathbb{R}^{J \times \prod_n I_n}$ to $\bigotimes_{i=1}^{N} \mathbf{A}^{(i)}$ with each $\mathbf{A}^{(i)} \in \mathbb{R}^{I_n \times R_n}$ is $\mathcal{O}(\sum_n I_n R_n + J \log J \sum_n R_n + J \log J \prod_n R_n)$. For simplicity, we will assume that $I_n = I$ and $R_n = R$ for all $n \in [N]$. We divide costs into costs per iteration of the main loop, and one-time costs due to computations outside the main loop.

### S3.2.1 TUCKER-ALS (algorithm given in main paper)

The dominant cost in TUCKER-ALS is the TTM product. Assuming that $I > NR$, the cost per iteration is $\mathcal{O}(RI^N)$. The final computation of $\mathcal{G}$ also costs $\mathcal{O}(RI^N)$. If $\mathcal{Y}$ is sparse these costs will be lower.

### S3.2.2 Algorithm S1: TUCKER-TS, multi-pass

---

**Algorithm S1:** TUCKER-TS, multi-pass (detailed)

---

**input** : $\mathcal{Y}$, $(R_1, R_2, \ldots, R_N)$, $(J_1, J_2)$
**output** : Rank-$(R_1, R_2, \ldots, R_N)$ Tucker decomposition $[\![\mathcal{G}; \mathbf{A}^{(1)}, \ldots, \mathbf{A}^{(N)}]\!]$ of $\mathcal{Y}$

1 Initialize $\mathcal{G}$, $\mathbf{A}^{(2)}, \mathbf{A}^{(3)}, \ldots, \mathbf{A}^{(N)}$
2 **repeat**
3     **for** $n = 1, \ldots, N$ **do**
4         For $k \in [N] \setminus n$, determine hash functions $h_k^{(1)} : [I_k] \to [J_1]$ and sign functions
        $s_k : [I_k] \to \{-1, +1\}$
5         $\mathbf{M}_1 = \text{TS}\left(\bigotimes_{\substack{i=N \\ i \neq n}}^{1} \mathbf{A}^{(i)}, \left\{h_i^{(1)}\right\}_{\substack{i=N \\ i \neq n}}^{1}, \{s_i\}_{\substack{i=N \\ i \neq n}}^{1}\right)$
6         $\mathbf{Y}_{(n), s_1}^{\top} = \text{TS}\left(\mathbf{Y}_{(n)}^{\top}, \left\{h_i^{(1)}\right\}_{\substack{i=N \\ i \neq n}}^{1}, \{s_i\}_{\substack{i=N \\ i \neq n}}^{1}\right)$
7         $\mathbf{A}^{(n)} = \arg\min_{\mathbf{A}} \left\|\mathbf{M}_1 \mathbf{G}_{(n)}^{\top} \mathbf{A}^{\top} - \mathbf{Y}_{(n), s_1}^{\top}\right\|_F^2$
8     **end**
9     For $k \in [N]$, determine hash functions $h_k^{(2)} : [I_k] \to [J_2]$ and sign functions
    $s_k : [I_k] \to \{-1, +1\}$
10     $\mathbf{M}_2 = \text{TS}\left(\bigotimes_{i=N}^{1} \mathbf{A}^{(i)}, \left\{h_i^{(2)}\right\}_{i=N}^{1}, \{s_i\}_{i=N}^{1}\right)$
11     $\mathbf{y}_{(:), s_2} = \text{TS}\left(\mathbf{y}_{(:)}, \left\{h_i^{(2)}\right\}_{i=N}^{1}, \{s_i\}_{i=N}^{1}\right)$
12     $\mathcal{G} = \arg\min_{\mathbf{z}} \left\|\mathbf{M}_2 \mathbf{z}_{(:)} - \mathbf{y}_{(:), s_2}\right\|_2^2$
13 **until** *termination criteria met*
14 **return** $\mathcal{G}$, $\mathbf{A}^{(1)}, \ldots, \mathbf{A}^{(N)}$

---

**One-time cost**    Negligible.

**Cost per iteration**    The inner for loop has the following costs:

- Line 5: We can TENSORSKETCH efficiently to a cost of $\mathcal{O}(NIRJ_1 + NRJ_1 \log J_1 + R^{N-1}J_1 \log J_1)$.

- Line 6: Since we have to compute this as a COUNTSKETCH, the cost is $\text{nnz}(\mathcal{Y})$.

- Line 7: The cost for computing $\mathbf{M}_1 \mathbf{G}_{(n)}^\top$ is $\mathcal{O}(J_1 R^N)$, the cost for QR factorizing this quantity is $\mathcal{O}(J_1 R^2)$, and the cost for solving each of the $I$ least-squares problems is $\mathcal{O}(I J_1 R + I R^2)$.

So the cost for the inner for loop is $\mathcal{O}(N^2 I R J_1 + N^2 R J_1 \log J_1 + N R^{N-1} J_1 \log J_1 + N \mathrm{nnz}(\mathcal{Y}) + N J_1 R^N + N I J_1 R + N I R^2)$. The cost of the remaining lines are:

- Line 10: We can TENSORSKETCH efficiently to a cost of $\mathcal{O}(N I R J_2 + N R J_2 \log J_2 + R^N J_2 \log J_2)$.
- Line 11: Since we have to compute this as a COUNTSKETCH, the cost is $\mathrm{nnz}(\mathcal{Y})$.
- Line 12: The cost for solving this system using e.g. QR factorization is $\mathcal{O}(J_2 R^{2N})$.

The total cost per iteration is therefore, to leading order,

$$\mathcal{O}(N \mathrm{nnz}(\mathcal{Y}) + N^2 I R J_1 + N^2 R J_1 \log J_1 + N R^{N-1} J_1 \log J_1 + N J_1 R^N \tag{S24}$$
$$+ N I J_1 R + N I R^2 + N I R J_2 + N R J_2 \log J_2 + R^N J_2 \log J_2 + J_2 R^{2N}). \tag{S25}$$

### S3.2.3 Algorithm S2: TUCKER-TS, one-pass with $I < J_1 + J_2$

---

**Algorithm S2:** TUCKER-TS, one-pass with all $I_n < J_1 + J_2$ (detailed)

---

**input** : $\mathcal{Y}$, $(R_1, R_2, \ldots, R_N)$, $(J_1, J_2)$
**output**: Rank-$(R_1, R_2, \ldots, R_N)$ Tucker decomposition $[\![\mathcal{G}; \mathbf{A}^{(1)}, \ldots, \mathbf{A}^{(N)}]\!]$ of $\mathcal{Y}$

1 For $n \in [N]$, determine hash functions $h_n^{(l)} : [I_n] \to [J_l]$ with $l = 1, 2$, and
   $s_n : [I_n] \to \{-1, +1\}$

2 Initialize $\mathcal{G}$. For $n \in [N]$, initialize $\mathbf{A}^{(n)}$ and compute $\hat{\mathbf{A}}_{s_1}^{(n)} = \left( \mathrm{FFT} \left( \mathrm{CS} \left( \mathbf{A}^{(n)}, h_n^{(1)}, s_n \right) \right) \right)^\top$

3 For $n \in [N]$, $\mathbf{Y}_{(n), s_1}^\top = \mathrm{TS} \left( \mathbf{Y}_{(n)}^\top, \left\{ h_i^{(1)} \right\}_{\substack{i=N \\ i \neq n}}^1, \{s_i\}_{\substack{i=N \\ i \neq n}}^1 \right)$

4 $\mathbf{y}_{(:), s_2} = \mathrm{TS} \left( \mathbf{y}_{(:)}, \left\{ h_i^{(2)} \right\}_{i=N}^1, \{s_i\}_{i=N}^1 \right)$

5 **repeat**

6      **for** $n = 1, \ldots, N$ **do**

7          $\mathbf{M}_1 = \mathrm{TS} \left( \left\{ \hat{\mathbf{A}}_{s_1}^{(i)} \right\}_{\substack{i=N \\ i \neq n}}^1 \right)$

8          $\mathbf{A}^{(n)} = \arg\min_{\mathbf{A}} \left\| \mathbf{M}_1 \mathbf{G}_{(n)}^\top \mathbf{A}^\top - \mathbf{Y}_{(n), s_1}^\top \right\|_F^2$

9          $\hat{\mathbf{A}}_{s_1}^{(n)} = \left( \mathrm{FFT} \left( \mathrm{CS} \left( \mathbf{A}^{(n)}, h_n^{(1)}, s_n \right) \right) \right)^\top$

10      **end**

11      $\mathbf{M}_2 = \mathrm{TS} \left( \bigotimes_{i=N}^1 \mathbf{A}^{(i)}, \left\{ h_i^{(2)} \right\}_{i=N}^1, \{s_i\}_{i=N}^1 \right)$

12      $\mathcal{G} = \arg\min_{\mathbf{z}} \left\| \mathbf{M}_2 \mathbf{z}_{(:)} - \mathbf{y}_{(:), s_2} \right\|_2^2$

13 **until** *termination criteria met*

14 **return** $\mathcal{G}$, $\mathbf{A}^{(1)}, \ldots, \mathbf{A}^{(N)}$

---

**One-time cost**

- Line 2: The cost of $N - 1$ COUNTSKETCHES and FFT of factor matrices is $\mathcal{O}(N I R + N R J_1 \log J_1)$.
- Lines 3 and 4: The sketching costs $\mathcal{O}(N \mathrm{nnz}(\mathcal{Y}))$.

So the total one-time cost is $\mathcal{O}(N \mathrm{nnz}(\mathcal{Y}) + N I R + N R J_1 \log J_1)$.

**Cost per iteration**    In the inner loop, we have the following costs:

- Line 7: The inverse FFT is the dominant cost at $\mathcal{O}(R^{N-1}J_1 \log J_1)$.
- Line 8: The cost for computing $\mathbf{M}_1 \mathbf{G}_{(n)}^\top$ is $\mathcal{O}(J_1 R^N)$, the cost for QR factorizing this quantity is $\mathcal{O}(J_1 R^2)$, and the cost for solving each of the $I$ least-squares problems is $\mathcal{O}(I J_1 R + I R^2)$.
- Line 9: The cost of a COUNTSKETCH and FFT is $\mathcal{O}(IR + R J_1 \log J_1)$.

So the total cost for the inner loop is $\mathcal{O}(N R^{N-1} J_1 \log J_1 + N J_1 R^N + N I J_1 R + N I R^2)$. We have the following additional costs per outer iteration:

- Line 11: We can TENSORSKETCH efficiently to a cost of $\mathcal{O}(N I R J_2 + N R J_2 \log J_2 + R^N J_2 \log J_2)$.
- Line 12: The cost for solving this system using e.g. QR factorization is $\mathcal{O}(J_2 R^{2N})$. However, since the design matrix in the normal equation formulation of this problem is usually well-conditioned, we instead choose to solve this step using conjugate gradient, at a cost $\mathcal{O}(R^{2N})$.

The total cost per iteration is therefore, to leading order,

$$\mathcal{O}(N R^{N-1} J_1 \log J_1 + N J_1 R^N + N I J_1 R + N I R^2 + N I R J_2 \tag{S26}$$
$$+ N R J_2 \log J_2 + R^N J_2 \log J_2 + R^{2N}). \tag{S27}$$

### S3.2.4   Algorithm S3: TUCKER-TS, one-pass with $I > J_1 + J_2$

**One-time costs**

- Line 2: The cost of $N-1$ COUNTSKETCHES and FFT of factor matrices is $\mathcal{O}(N I R + N R J_1 \log J_1)$.
- Line 3: The TENSORSKETCH costs $\mathcal{O}(N \text{nnz}(\mathcal{Y}))$ and the subsequent smaller COUNTSKETCHES cost $\mathcal{O}(N \min(I J_1, \text{nnz}(\mathcal{Y})))$.
- Line 4: Since we have to compute this as a COUNTSKETCH, the cost is $\text{nnz}(\mathcal{Y})$.

So the upfront cost is $\mathcal{O}(N \text{nnz}(\mathcal{Y}) + N I R + N R J_1 \log J_1)$, but with some additional smaller computations compared to Algorithm S2. We will also have some additional one-time costs after the end of the main loop:

- Line 16: The inverse FFT is the dominant cost at $\mathcal{O}(N R^{N-1} J_1 \log J_1)$ for $N$ repetitions.
- Line 17: The cost for computing $\mathbf{M}_1 \mathbf{G}_{(n)}^\top$ is $\mathcal{O}(J_1 R^N)$, the cost for QR factorizing this quantity is $\mathcal{O}(J_1 R^2)$, and the cost for solving each of the $I$ least-squares problems is $\mathcal{O}(I J_1 R + I R^2)$. So the dominant cost, for $N$ repetitions, is $\mathcal{O}(N J_1 R^N + N I J_1 R + N I R^2)$.
- Line 19: The cost of the COUNTSKETCH and the FFT is $\mathcal{O}(N I R + N R J_1 \log J_1)$ for $N-1$ repetitions.
- Line 22: The cost of applying TENSORSKETCH is $\mathcal{O}(N I R J_2 + N R J_2 \log J_2 + R^N J_2 \log J_2)$.
- Line 23: The cost for solving this system using e.g. QR factorization is $\mathcal{O}(J_2 R^{2N})$.

The total one-time cost is therefore

$$\mathcal{O}(N \text{nnz}(\mathcal{Y}) + N R^{N-1} J_1 \log J_1 + N J_1 R^N + N I J_1 R + N I R^2 + N I R J_2 \tag{S28}$$
$$+ N R J_2 \log J_2 + R^N J_2 \log J_2 + J_2 R^{2N}). \tag{S29}$$

**Cost per iteration**

- Line 7: The inverse FFT is the dominant cost at $\mathcal{O}(R^{N-1} J_1 \log J_1)$.
- Line 8: The cost for computing $\mathbf{M}_1 \mathbf{G}_{(n)}^\top$ is $\mathcal{O}(J_1 R^N)$, the cost for QR factorizing this quantity is $\mathcal{O}(J_1 R^2)$, and the cost for solving each of the $J_1 + J_2$ least-squares problems is $\mathcal{O}(J_1^2 R + J_1 R^2 + J_1 J_2 R + J_2 R^2)$. The leading order cost for the line is $\mathcal{O}(J_1 R^N + J_1^2 R + J_1 J_2 R + J_2 R^2)$.

**Algorithm S3:** TUCKER-TS, one-pass with all $I_n > J_1 + J_2$ (detailed)

---

**input** : $\mathcal{Y}$, $(R_1, R_2, \ldots, R_N)$, $(J_1, J_2)$
**output** : Rank-$(R_1, R_2, \ldots, R_N)$ Tucker decomposition $[\![\mathcal{G}; \mathbf{A}^{(1)}, \ldots, \mathbf{A}^{(N)}]\!]$ of $\mathcal{Y}$

**1** For $n \in [N]$, determine hash functions $h_n^{(l)} : [I_n] \to [J_l]$ with $l = 1, 2$, and
  $s_n : [I_n] \to \{-1, +1\}$

**2** Initialize $\mathcal{G}$. For $n \in [N]$, initialize $\mathbf{A}^{(n)}$ and compute $\hat{\mathbf{A}}_{s_1}^{(n)} = \left( \text{FFT} \left( \text{CS} \left( \mathbf{A}^{(n)}, h_n^{(1)}, s_n \right) \right) \right)^{\top}$

**3** For $n \in [N]$, $\mathbf{Y}_{(n),s_1}^{\top} = \text{TS} \left( \mathbf{Y}_{(n)}^{\top}, \left\{ h_i^{(1)} \right\}_{\substack{i=N \\ i \neq n}}^{1}, \{s_i\}_{\substack{i=N \\ i \neq n}}^{1} \right)$, $\mathbf{Y}_{(n),s_1'} = \text{CS} \left( \mathbf{Y}_{(n),s_1}, h_n^{(1)}, s_n \right)$,

  $\mathbf{Y}_{(n),s_1''} = \text{CS} \left( \mathbf{Y}_{(n),s_1}, h_n^{(2)}, s_n \right)$

**4** $\mathbf{y}_{(:),s_2} = \text{TS} \left( \mathbf{Y}_{(:)}, \left\{ h_i^{(2)} \right\}_{i=N}^{1}, \{s_i\}_{i=N}^{1} \right)$

**5 repeat**

**6**     **for** $n = 1, \ldots, N$ **do**

**7**        $\mathbf{M}_1 = \text{TS} \left( \left\{ \hat{\mathbf{A}}_{s_1}^{(i)} \right\}_{\substack{i=N \\ i \neq n}}^{1} \right)$

**8**        $[\mathbf{A}_{s_1}, \mathbf{A}_{s_2}] = \arg\min_{\mathbf{A}} \left\| \mathbf{M}_1 \mathbf{G}_{(n)}^{\top} \mathbf{A}^{\top} - \left[ \mathbf{Y}_{(n),s_1'}^{\top}, \mathbf{Y}_{(n),s_1''}^{\top} \right] \right\|_F^2$

**9**        $\hat{\mathbf{A}}_{s_1}^{(n)} = \left( \text{FFT} \left( \mathbf{A}_{s_1} \right) \right)^{\top}$

**10**        $\hat{\mathbf{A}}_{s_2}^{(n)} = \left( \text{FFT} \left( \mathbf{A}_{s_2} \right) \right)^{\top}$

**11**     **end**

**12**     $\mathbf{M}_2 = \text{FFT}^{-1} \left( \left( \bigodot_{i=N}^{1} \hat{\mathbf{A}}_{s_2}^{(i)} \right)^{\top} \right)$

**13**     $\mathcal{G} = \arg\min_{\mathbf{z}} \left\| \mathbf{M}_2 \mathbf{z}_{(:)} - \mathbf{y}_{(:),s_2} \right\|_2^2$

**14 until** *termination criteria met*

**15 for** $n = 1, \ldots, N$ **do**

**16**     $\mathbf{M}_1 = \text{TS} \left( \left\{ \hat{\mathbf{A}}_{s_1}^{(i)} \right\}_{\substack{i=N \\ i \neq n}}^{1} \right)$

**17**     $\mathbf{A}^{(n)} = \arg\min_{\mathbf{A}} \left\| \mathbf{M}_1 \mathbf{G}_{(n)}^{\top} \mathbf{A}^{\top} - \mathbf{Y}_{(n),s_1}^{\top} \right\|_F^2$

**18**     **if** $n < N$ **then**

**19**        $\hat{\mathbf{A}}_{s_1}^{(n)} = \left( \text{FFT} \left( \text{CS} \left( \mathbf{A}^{(n)}, h_n^{(1)}, s_n \right) \right) \right)^{\top}$

**20**     **end**

**21 end**

**22** $\mathbf{M}_2 = \text{TS} \left( \bigotimes_{i=N}^{1} \mathbf{A}^{(i)}, \left\{ h_i^{(2)} \right\}_{i=N}^{1}, \{s_i\}_{i=N}^{1} \right)$

**23** $\mathcal{G} = \arg\min_{\mathbf{z}} \left\| \mathbf{M}_2 \mathbf{z}_{(:)} - \mathbf{y}_{(:),s_2} \right\|_2^2$

**24 return** $\mathcal{G}$, $\mathbf{A}^{(1)}, \ldots, \mathbf{A}^{(N)}$

---

- Lines 9 and 10: The cost of these two lines is $\mathcal{O}(RJ_1 \log J_1 + RJ_2 \log J_2)$.

So the total cost for the inner for loop is therefore $\mathcal{O}(NR^{N-1}J_1 \log J_1 + NJ_1 R^N + NJ_1^2 R + NJ_1 J_2 R + NJ_2 R^2 + NRJ_2 \log J_2)$. The cost of the remaining lines are:

- Line 12: The inverse FFT is the dominant cost at $\mathcal{O}(R^N J_2 \log J_2)$.

- Line 13: The cost of solving this system e.g. via QR factorization is $\mathcal{O}(J_2 R^{2N})$.

The total cost per iteration is therefore

$$\mathcal{O}(NR^{N-1}J_1 \log J_1 + NJ_1R^N + NJ_1^2R + NJ_1J_2R \tag{S30}$$

$$+ NJ_2R^2 + NRJ_2 \log J_2 + R^N J_2 \log J_2 + J_2 R^{2N}). \tag{S31}$$

### S3.2.5   Algorithm S4: TUCKER-TTMTS

---

**Algorithm S4:** TUCKER-TTMTS (detailed)

---

**input** : $\mathcal{Y}$, $(R_1, R_2, \ldots, R_N)$, $(J_1, J_2)$
**output** : Rank-$(R_1, R_2, \ldots, R_N)$ Tucker decomposition $[\![\mathcal{G}; \mathbf{A}^{(1)}, \ldots, \mathbf{A}^{(N)}]\!]$ of $\mathcal{Y}$

1  For $n \in [N]$, determine hash functions $h_n^{(l)} : [I_n] \to [J_l]$ with $l = 1, 2$, and
   $s_n : [I_n] \to \{-1, +1\}$

2  Initialize $\mathcal{G}$. For $n \in [N]$, initialize $\mathbf{A}^{(n)}$ and compute $\hat{\mathbf{A}}_{s_1}^{(n)} = \left( \text{FFT} \left( \text{CS} \left( \mathbf{A}^{(n)}, h_n^{(1)}, s_n \right) \right) \right)^\top$

3  For $n \in [N]$, $\mathbf{Y}_{(n), s_1}^\top = \text{TS} \left( \mathbf{Y}_{(n)}^\top, \left\{ h_i^{(1)} \right\}_{\substack{i=N \\ i \neq n}}^1, \{s_i\}_{\substack{i=N \\ i \neq n}}^1 \right)$

4  $\mathbf{y}_{(:), s_2} = \text{TS} \left( \mathbf{y}_{(:)}, \left\{ h_i^{(2)} \right\}_{i=N}^1, \{s_i\}_{i=N}^1 \right)$

5  **repeat**
6      **for** $n = 1, \ldots, N$ **do**
7          $\mathbf{M}_1 = \text{TS} \left( \left\{ \hat{\mathbf{A}}_{s_1}^{(i)} \right\}_{\substack{i=N \\ i \neq n}}^1 \right)$
8          $\mathbf{Z}_{(n)} = \mathbf{Y}_{(n), s_1} \mathbf{M}_1$
9          $\mathbf{A}^{(n)} = R_n$ leading left singular vectors of $\mathbf{Z}_{(n)}$
10     **end**
11 **until** *termination criteria met*
12 $\mathbf{M}_2 = \text{TS} \left( \bigotimes_{i=N}^1 \mathbf{A}^{(i)}, \left\{ h_i^{(2)} \right\}_{i=N}^1, \{s_i\}_{i=N}^1 \right)$
13 $\mathbf{g}_{(:)} = \mathbf{M}_2^\top \mathbf{y}_{(:), s_2}$
14 **return** $\mathcal{G}$, $\mathbf{A}^{(1)}, \ldots, \mathbf{A}^{(N)}$

---

**One-time costs**

- Line 2: The cost of $N - 1$ COUNTSKETCHES and FFTs of factor matrices is $\mathcal{O}(NIR + NRJ_1 \log J_1)$.
- Lines 3 and 4: The sketching costs $\mathcal{O}(N\text{nnz}(\mathcal{Y}))$.
- Line 12: We can TENSORSKETCH efficiently to a cost of $\mathcal{O}(NIRJ_2 + NRJ_2 \log J_2 + R^N J_2 \log J_2)$.
- Line 13: The cost of this matrix-vector multiplication is $\mathcal{O}(J_2 R^N)$.

The total one-time cost is therefore $\mathcal{O}(NIRJ_2 + N\text{nnz}(\mathcal{Y}) + NRJ_1 \log(J_1) + NRJ_2 \log(J_2) + R^N J_2 \log(J_2))$.

**Cost per iteration**

- Line 7: The inverse FFT is the dominant cost at $\mathcal{O}(R^{N-1}J_1 \log J_1)$.
- Line 8: The cost of this matrix-matrix multiplication is $\mathcal{O}(J_1 I R^{N-1})$.
- Line 9: We use the randomized SVD of [5], the cost of which is $\mathcal{O}(IR^{N-1} \log(R) + R^{N+1})$

Each of these are repeated $N$ times per outer iteration, so the total cost per outer iteration is therefore
$\mathcal{O}(NR^{N-1}J_1 \log(J_1) + NJ_1 I R^{N-1} + NIR^{N-1} \log(R) + NR^{N+1})$.

### S3.2.6 Summary of complexity results

In Tables S1–S4 we summarize the results above. For clarity, we have made the additional assumption that $J_1 = KR^{N-1}$ and $J_2 = KR^N$ for some constant $K > 1$. We have found that this choice of $J_1$ and $J_2$ often works well in practice. Tables S1 and S2 give the leading order complexities for one-time costs and per iteration costs, respectively. Tables S3 and S4 give the corresponding complexities for each of the cases when the variables $I$, $R$ and $K$ are assumed to be large.

| Alg. | One-time cost |
|------|---------------|
| T.-ALS | $RI^N$ |
| S1 | Negligible |
| S2 | $N\mathrm{nnz}(\boldsymbol{\mathcal{Y}}) + NIR + NKR^N \log K + N^2 KR^N \log R$ |
| S3 | $N\mathrm{nnz}(\boldsymbol{\mathcal{Y}}) + NR^{2N-2}K \log K + N^2 KR^{2N-2} \log R + NKR^{2N-1} + NIKR^{N+1} + R^{2N}K \log K + NKR^{2N} \log R + KR^{3N}$ |
| S4 | $NIKR^{N+1} + N\mathrm{nnz}(\boldsymbol{\mathcal{Y}}) + NKR^{N+1} \log K + N^2 KR^{N+1} \log R + KR^{2N} \log K + KNR^{2N} \log R$ |

Table S1: Comparison of computational complexity for one-time computations for TUCKER-ALS and Algorithms S1–S4. All complexities are to leading order.

| Alg. | Per iteration cost |
|------|--------------------|
| T.-ALS | $RI^N$ |
| S1 | $N\mathrm{nnz}(\boldsymbol{\mathcal{Y}}) + N^2 IKR^N + N^2 KR^N \log K + N^3 KR^N \log R + NKR^{2N-2} \log K + N^2 KR^{2N-2} \log R + NIKR^{N+1} + KR^{2N} \log K + NKR^{2N} \log R + KR^{3N}$ |
| S2 | $NR^{2N-2}K \log K + N^2 KR^{2N-2} \log R + NKR^{2N-1} + NIKR^{N+1} + R^{2N}K \log K + NKR^{2N} \log R$ |
| S3 | $NR^{2N-2}K \log K + N^2 KR^{2N-2} \log R + NK^2 R^{2N} + R^{2N}K \log K + NKR^{2N} \log R + KR^{3N}$ |
| S4 | $NKR^{2N-2} \log K + KN^2 R^{2N-2} \log R + NKIR^{2N-2}$ |

Table S2: Comparison of computational complexity per iteration for TUCKER-ALS and Algorithms S1–S4. All complexities are to leading order.

| | One-time cost when the given variable is large | | |
|------|-----|-----|-----|
| Alg. | $I$ | $R$ | $K$ |
| T.-ALS | $RI^N$ | $RI^N$ | n/a |
| S1 | Negligible | Negligible | Negligible |
| S2 | $N\mathrm{nnz}(\boldsymbol{\mathcal{Y}}) + NIR$ | $N^2 KR^N \log R$ | $NR^N K \log K$ |
| S3 | $N\mathrm{nnz}(\boldsymbol{\mathcal{Y}}) + NIKR^{N+1}$ | $KR^{3N}$ | $(NR^{2N-2} + R^{2N}) \times K \log K$ |
| S4 | $N\mathrm{nnz}(\boldsymbol{\mathcal{Y}}) + NIKR^{N+1}$ | $KNR^{2N} \log R$ | $(NR^{N+1} + R^{2N})K \log K$ |

Table S3: Comparison of computational complexity for one-time computations for TUCKER-ALS and Algorithms S1–S4 when each of the variables $I$, $R$ and $K$ are assumed to be large. All complexities are to leading order.

## S4 Proof of Proposition 3.1

Let $V$ be the subspace spanned by the columns of $\mathbf{U} \stackrel{\text{def}}{=} \bigotimes_{i=N}^{1} \mathbf{A}^{(i)}$. Note that since each $\mathbf{A}^{(i)}$ has orthonormal columns, so does $\mathbf{U}$ [10]. Suppose $J_2 \geq (\prod_n R_n)^2 (2 + 3^N)/(\varepsilon^2 \delta)$. Using

| | Per iteration cost when the given variable is large | | |
|---|---|---|---|
| Alg. | $I$ | $R$ | $K$ |
| T.-ALS | $RI^N$ | $RI^N$ | n/a |
| S1 | $N\mathrm{nnz}(\mathcal{Y}) + N^2 IKR^N$ $+ NIKR^{N+1}$ | $KR^{3N}$ | $(N^2 R^N + NR^{2N-2} + R^{2N})$ $\times K \log K$ |
| S2 | $NIKR^{N+1}$ | $NKR^{2N} \log R$ | $NR^{2N-2} K \log K$ $+ R^{2N} K \log K$ |
| S3 | Negligible | $KR^{3N}$ | $NK^2 R^{2N}$ |
| S4 | $NIKR^{2N-2}$ | $KN^2 R^{2N-2} \log R$ | $NR^{2N-2} K \log K$ |

Table S4: Comparison of computational complexity per iteration for TUCKER-ALS and Algorithms S1–S4 when each of the variables $I$, $R$ and $K$ are assumed to be large. All complexities are to leading order.

Theorem 4.2.2 in [6], we have that the maximum and minimum eigenvalues of $\mathbf{M}$ satisfy

$$\lambda_{\max}(\mathbf{M}) = \max_{\mathbf{x}:\|\mathbf{x}\|=1} \left\| \mathbf{T}^{(N+1)} \mathbf{U}\mathbf{x} \right\|_2^2, \tag{S32}$$

$$\lambda_{\min}(\mathbf{M}) = \min_{\mathbf{x}:\|\mathbf{x}\|=1} \left\| \mathbf{T}^{(N+1)} \mathbf{U}\mathbf{x} \right\|_2^2. \tag{S33}$$

$$\tag{S34}$$

Due to Lemma B.1 in [3], we therefore have

$$\lambda_{\max}(\mathbf{M}) \leq (1+\varepsilon)^2 \max_{\mathbf{x}:\|\mathbf{x}\|=1} \|\mathbf{U}\mathbf{x}\|_2^2 = (1+\varepsilon)^2 \max_{\mathbf{x}:\|\mathbf{x}\|=1} \|\mathbf{x}\|_2^2 = (1+\varepsilon)^2, \tag{S35}$$

$$\lambda_{\min}(\mathbf{M}) \geq (1-\varepsilon)^2 \min_{\mathbf{x}:\|\mathbf{x}\|=1} \|\mathbf{U}\mathbf{x}\|_2^2 = (1-\varepsilon)^2 \min_{\mathbf{x}:\|\mathbf{x}\|=1} \|\mathbf{x}\|_2^2 = (1-\varepsilon)^2, \tag{S36}$$

$$\tag{S37}$$

with probability at least $1 - \delta$, since $\mathbf{U}\mathbf{x} \in V$ for all $\mathbf{x} \in \mathbb{R}^{\Pi_n R_n}$. Since $\mathbf{M}$ is symmetric and positive semi-definite, we also have singular values $\sigma_{\max}(\mathbf{M}) = \lambda_{\max}(\mathbf{M}) \leq (1+\varepsilon)^2$, $\sigma_{\min}(\mathbf{M}) = \lambda_{\min}(\mathbf{M}) \geq (1-\varepsilon)^2$. The result now follows:

$$\kappa(\mathbf{M}) = \frac{\sigma_{\max}}{\sigma_{\min}} \leq \frac{(1+\varepsilon)^2}{(1-\varepsilon)^2}. \tag{S38}$$

## S5 Formal statement and proof of Proposition 3.3

Let

$$\mathcal{G} = \mathcal{Y} \times_1 \mathbf{A}^{(1)\top} \times_2 \mathbf{A}^{(2)\top} \cdots \times_N \mathbf{A}^{(N)\top}. \tag{S39}$$

Moreover, let $\mathbf{A}^*$ and $\mathcal{G}^*$ be the updated values on lines 5 and 8 in Algorithm 1 in the main manuscript, and let $\tilde{\mathbf{A}}$ and $\tilde{\mathcal{G}}$ be the corresponding updates in Algorithm 3 in the main manuscript.

**Proposition S5.1** (TUCKER-TTMTS). *Assume that* $\mathbf{G}_{(n)}\mathbf{G}_{(n)}^{\top}$ *is invertible, with* $\mathcal{G}$ *given in* (S39), *and that each* TENSORSKETCH *operator is redefined prior to being used. If the sketch dimension* $J_1 \geq (2 + 3^{N-1})/(\varepsilon^2 \delta)$, *then with probability at least* $1 - \delta$ *it holds that*

$$\left\| \left( \bigotimes_{\substack{i=N \\ i \neq n}}^{1} \mathbf{A}^{(i)} \right) \mathbf{G}_{(n)}^{\top} \tilde{\mathbf{A}}^{\top} - \mathbf{Y}_{(n)}^{\top} \right\|_F \leq \left\| \left( \bigotimes_{\substack{i=N \\ i \neq n}}^{1} \mathbf{A}^{(i)} \right) \mathbf{G}_{(n)}^{\top} \mathbf{A}^{*\top} - \mathbf{Y}_{(n)}^{\top} \right\|_F \tag{S40}$$

$$+ \varepsilon \left\| \left( \mathbf{G}_{(n)} \mathbf{G}_{(n)}^{\top} \right)^{-1} \right\|_F \|\mathcal{G}\|^2 \|\mathcal{Y}\| \prod_{\substack{i=1 \\ i \neq n}}^{N} R_i. \tag{S41}$$

*Moreover, if the sketch dimension $J_2 \geq (2 + 3^N)/(\varepsilon^2 \delta)$, then with probability at least $1 - \delta$ it holds that*

$$\left\| \left( \bigotimes_{i=N}^{1} \mathbf{A}^{(i)} \right) \tilde{\mathbf{g}}_{(:)} - \mathbf{y}_{(:)} \right\|_2 \leq \left\| \left( \bigotimes_{i=N}^{1} \mathbf{A}^{(i)} \right) \mathbf{g}^*_{(:)} - \mathbf{y}_{(:)} \right\|_2 + \varepsilon \left\| \mathbf{\mathcal{Y}} \right\| \sqrt{\prod_{i=1}^{N} R_i}. \tag{S42}$$

*Proof.* It is straightforward to show that $\mathbf{A}^*$ solves

$$\mathbf{A}^* = \underset{\mathbf{A} \in \mathbb{R}^{I_n \times R_n}}{\arg\min} \left\| \left( \bigotimes_{\substack{i=N \\ i \neq n}}^{1} \mathbf{A}^{(i)} \right) \mathbf{G}_{(n)}^\top \mathbf{A}^\top - \mathbf{X}_{(n)}^\top \right\|_F^2 ; \tag{S43}$$

see e.g. the discussion in Subsection 4.2 in [7]. Setting the gradient of the objective function equal to zero and rearranging, we get that

$$\mathbf{G}_{(n)} \mathbf{G}_{(n)}^\top \mathbf{A}^{*\top} = \mathbf{G}_{(n)} \left( \bigotimes_{\substack{i=N \\ i \neq n}}^{1} \mathbf{A}^{(i)} \right)^\top \mathbf{Y}_{(n)}^\top. \tag{S44}$$

By a similar argument, we find

$$\mathbf{G}_{(n)} \mathbf{G}_{(n)}^\top \tilde{\mathbf{A}}^\top = \mathbf{G}_{(n)} \left( \bigotimes_{\substack{i=N \\ i \neq n}}^{1} \mathbf{A}^{(i)} \right)^\top \mathbf{T}^{(n)\top} \mathbf{T}^{(n)} \mathbf{Y}_{(n)}^\top. \tag{S45}$$

The matrix $\tilde{\mathbf{A}}^\top$ is therefore the solution to a perturbed version of the system in (S44). Using Lemma B.1 in [3], we get that the perturbation satisfies

$$\left\| \mathbf{G}_{(n)} \left( \bigotimes_{\substack{i=N \\ i \neq n}}^{1} \mathbf{A}^{(i)} \right)^\top \mathbf{Y}_{(n)}^\top - \mathbf{G}_{(n)} \left( \bigotimes_{\substack{i=N \\ i \neq n}}^{1} \mathbf{A}^{(i)} \right)^\top \mathbf{T}^{(n)\top} \mathbf{T}^{(n)} \mathbf{Y}_{(n)}^\top \right\|_F \tag{S46}$$

$$\leq \varepsilon \left\| \mathbf{\mathcal{G}} \right\| \left\| \bigotimes_{\substack{i=N \\ i \neq n}}^{1} \mathbf{A}^{(i)} \right\|_F \left\| \mathbf{\mathcal{Y}} \right\| \tag{S47}$$

with probability at least $1 - \delta$ if $J_1 \geq (2 + 3^{N-1})/(\varepsilon^2 \delta)$. Standard sensitivity analysis in linear algebra (see e.g. Subsection 2.6 in [4]) now gives that

$$\left\| \mathbf{A}^{*\top} - \tilde{\mathbf{A}}^\top \right\|_F \leq \varepsilon \left\| \left( \mathbf{G}_{(n)} \mathbf{G}_{(n)}^\top \right)^{-1} \right\|_F \left\| \mathbf{\mathcal{G}} \right\| \left\| \bigotimes_{\substack{i=N \\ i \neq n}}^{1} \mathbf{A}^{(i)} \right\|_F \left\| \mathbf{\mathcal{Y}} \right\| \tag{S48}$$

with probability at least $1 - \delta$ if $J_1 \geq (2 + 3^{N-1})/(\varepsilon^2 \delta)$. It follows that

$$\left\| \left( \bigotimes_{\substack{i=N \\ i \neq n}}^{1} \mathbf{A}^{(i)} \right) \mathbf{G}_{(n)}^\top \tilde{\mathbf{A}}^\top - \mathbf{Y}_{(n)}^\top \right\|_F \tag{S49}$$

$$\leq \left\| \left( \bigotimes_{\substack{i=N \\ i \neq n}}^{1} \mathbf{A}^{(i)} \right) \mathbf{G}_{(n)}^\top \mathbf{A}^{*\top} - \mathbf{Y}_{(n)}^\top \right\|_F + \left\| \bigotimes_{\substack{i=N \\ i \neq n}}^{1} \mathbf{A}^{(i)} \right\|_F \left\| \mathbf{\mathcal{G}} \right\| \left\| \mathbf{A}^{*\top} - \tilde{\mathbf{A}}^\top \right\|_F \tag{S50}$$

$$\leq \left\| \left( \bigotimes_{\substack{i=N \\ i \neq n}}^{1} \mathbf{A}^{(i)} \right) \mathbf{G}_{(n)}^\top \mathbf{A}^{*\top} - \mathbf{Y}_{(n)}^\top \right\|_F + \varepsilon \left\| \left( \mathbf{G}_{(n)} \mathbf{G}_{(n)}^\top \right)^{-1} \right\|_F \left\| \mathbf{\mathcal{G}} \right\|^2 \left\| \mathbf{\mathcal{Y}} \right\| \prod_{\substack{i=1 \\ i \neq n}}^{N} R_i \tag{S51}$$

$$\tag{S52}$$

with probability at least $1 - \delta$ if $J_1 \geq (2 + 3^{N-1})/(\varepsilon^2 \delta)$, and where we used the fact that $\bigotimes_{\substack{i=N \\ i \neq n}}^{1} \mathbf{A}^{(i)}$ is an orthogonal matrix of size $(\prod_{i \neq n} I_i) \times (\prod_{i \neq n} R_i)$ and therefore

$$
\left\| \bigotimes_{\substack{i=N \\ i \neq n}}^{1} \mathbf{A}^{(i)} \right\|_F^2 = \prod_{\substack{i=1 \\ i \neq n}}^{N} R_i. \tag{S53}
$$

For the other claim, Lemma B.1 in [3] gives that

$$
\left\| \mathbf{g}_{(:)}^* - \tilde{\mathbf{g}}_{(:)} \right\|_2 = \left\| \left( \bigotimes_{i=N}^{1} \mathbf{A}^{(i)} \right)^\top \mathbf{y}_{(:)} - \left( \mathbf{T}^{(N+1)} \bigotimes_{i=N}^{1} \mathbf{A}^{(i)} \right)^\top \mathbf{T}^{(N+1)} \mathbf{y}_{(:)} \right\|_F \tag{S54}
$$

$$
\leq \varepsilon \left\| \bigotimes_{i=N}^{1} \mathbf{A}^{(i)} \right\|_F \| \boldsymbol{\mathcal{Y}} \| \tag{S55}
$$

with probability at least $1 - \delta$ if $J_2 \geq (2 + 3^N)/(\varepsilon^2 \delta)$. It follows that

$$
\left\| \left( \bigotimes_{i=N}^{1} \mathbf{A}^{(i)} \right) \tilde{\mathbf{g}}_{(:)} - \mathbf{y}_{(:)} \right\|_2 \tag{S56}
$$

$$
\leq \left\| \left( \bigotimes_{i=N}^{1} \mathbf{A}^{(i)} \right) \mathbf{g}_{(:)}^* - \mathbf{y}_{(:)} \right\|_2 + \left\| \bigotimes_{i=N}^{1} \mathbf{A}^{(i)} \right\|_2 \left\| \mathbf{g}_{(:)}^* - \tilde{\mathbf{g}}_{(:)} \right\|_2 \tag{S57}
$$

$$
\leq \left\| \left( \bigotimes_{i=N}^{1} \mathbf{A}^{(i)} \right) \mathbf{g}_{(:)}^* - \mathbf{y}_{(:)} \right\|_2 + \varepsilon \| \boldsymbol{\mathcal{Y}} \| \sqrt{\prod_{i=1}^{N} R_i}, \tag{S58}
$$

where we again use the fact that $\bigotimes_{i=N}^{1} \mathbf{A}^{(i)}$ is a $(\prod_i I_i) \times (\prod_i R_i)$ orthogonal matrix, and its 2-norm therefore is 1, and its Frobenius norm is equal to $\sqrt{\prod_i R_i}$. □

## S6 Further experiments

In this section, we present some further experiments. Figures S1 and S2 shows additional examples of how the error of TUCKER-TS and TUCKER-TTMTS, relative to that of TUCKER-ALS, changes with the sketch dimension parameter $K$. The figures also show how the algorithms perform when each TENSORSKETCH operator is redefined prior to being used (called "multi-pass" in the figures). We note that our approach of defining the TENSORSKETCH operators upfront consistently leads to a lower error compared to when the sketch operators are redefined before being used each time. Figures S3 and S4 show how the algorithms scale with increased dimension size $I$ for 4-way tensors. TUCKER-ALS, MET and MACH run out of memory when $I = 1e+5$. For reasons explained in the main manuscript, FSTD1 does not work well on tensors that are very sparse, which we can observe here.

Figure S1: Further examples of how the error of TUCKER-TS, relative to that of TUCKER-ALS, is impacted by the sketch dimension parameter $K$. For plot (a), the tensor size is $500 \times 500 \times 500$ with $\text{nnz}(\mathcal{Y}) \approx$ 1e+6, and with both the true and algorithm target rank equal to $(10, 10, 10)$. For plot (b), the tensor size is $100 \times 100 \times 100 \times 100$ with $\text{nnz}(\mathcal{Y}) \approx$ 1e+7, and with both the true and algorithm target rank equal to $(5, 5, 5, 5)$. For plot (c), the tensor size is $100 \times 100 \times 100 \times 100$ with $\text{nnz}(\mathcal{Y}) \approx$ 1e+7, with true rank $(10, 10, 10, 10)$ and algorithm target rank $(5, 5, 5, 5)$. For plot (d), the tensor is dense and of size $500 \times 500 \times 500$, and with both the true and algorithm target rank equal to $(10, 10, 10)$. For plot (e), the tensor is dense and of size $500 \times 500 \times 500$, and with true rank $(15, 15, 15)$ and algorithm target rank $(10, 10, 10)$.

Figure S2: Further examples of how the error of TUCKER-TTMTS, relative to that of TUCKER-ALS, is impacted by the sketch dimension parameter $K$. The tensor properties for each subplot are the same as in Figure S1.

Figure S3: Comparison of relative error and run time for random sparse 4-way tensors. The number of nonzero elements is nnz($\mathcal{Y}$) $\approx$ 1e+6. Both the true and target ranks are $(5, 5, 5, 5)$. We used $K = 10$ in these experiments.

Figure S4: Comparison of relative error and run time for random sparse 4-way tensors. The number of nonzero elements is nnz($\mathcal{Y}$) $\approx$ 1e+6. The true rank is $(7, 7, 7, 7)$ and the algorithm target rank is $(5, 5, 5, 5)$. In these experiments, we used $K = 10$ and a convergence tolerance of 1e-1.