[Reviews · NeurIPS 2018]

Reviewer 1



In this paper, the authors present two variants of a random projection-based algorithm (TensorSketch) whose purpose is to obtain a low-rank Tucker approximation for a given multidimensional dataset. It includes the possibility that entries of the data tensor are being streamed in one specific mode. I found the topic interesting and relevant to NIPS because this kind of approach is very important for developing real world big data applications with multidimensional datasets. However, I found the paper not well structured and lacking of clarity in the presentation among other issues. The authors recognized these issues in their response, found my comments useful to improve the quality of the paper and said they will incorporate changes in the final version. However, there is not detailed information about how the issues will be solved. The main issues I found are the following: - An introduction to “random sketching” is missing. The authors start using the term in the introduction but the it is not explained throughout the paper. They just limited themselves to give references to previous papers. Authors said, they don't want to repeat work from others. - Section 1.2 "A brief introduction to TensorSketch" is very confusing. The authors refer to previous papers where this technique was proposed but it is not explained so the reader gets lost without reading previous works. There are additional information about TensorSketch in the Supplementary information but I found it not useful and very disconnected from the main text of the article. - The usage of matrix B in lines 73 – 76 is misleading. I think there is a mistake and it should be replaced by A in lines 73 and 76. - In line 61, the method COUNTSKETCH is introduced but never explained which makes difficult to follow the article. - Equation (2) is not well motivated and its introduction in section 1.2 is confusing. There is some information about this formula in the Supplemental info but there is no link in the main text to the explanations. Moreover, the operator S^{(n)} is not defined in section 1.2. - First part of section 3, lines 113 – 122, including the classical algorithm HOOI (Algorithm 1: Tucker ALS) should be moved to the introduction section 1.1. - Section 3.1 (Tucker-TS) is not clearly described. For example, why two operators S_1 and S_2 are needed? How dimension J1 and J2 are chosen? - Authors say that, TensorSketch algorithm meet a theoretical condition of [10] but it is not explained here. Also, they say that avoiding defining a new sketch at every iteration showed better performance in the experiments (Fig. 1) but they were not able to explain this behavior or give some interpretation of this result. - Propositions 3.1 and 3.3 are difficult to interpret without some additional explanations - In Algorithm 2:  Why two parameters J1 and J2 are needed? How these parameters are chosen in practice? Authors propose to have J1=K*R^(N-1) and J2=K*R^N as a practical choice, however it is not clear how to select an optimal value of parameter K.  Only sizes of operators T^(n) are defined in line 2 but how these matrices are generated? Are they Gaussian i.i.d matrices? This should be clearly presented and justified. These questions were not addressed by the authors in their responses. - Algorithm of section 3.2 is not well motivated. Why is this algorithm proposed? - In the Complexity analysis, the proposed algorithms are compared only to the classical Tucker-ALS algorithm. I think they should compare also to other algorithms such as the ones used in the Experiments. For example, FSTD1 algorithm paper provided a detailed analysis of the complexity. - Figs. 2-3, 4-5 show comparison with other algorithms for sparse tensors. The results of FSTD1 and MACH show relative errors very close to 1 (100%), which, as I understand correspond to totally wrong approximations. The authors justify the poor performance of FSTD1 algorithm because of the sparsity of the tensor. I think this comparison are not useful because FSTD1 is not well defined for sparse tensors. I think comparison of algorithms should be performed also on dense tensors, as the one included in section 4.2. - In section 4.2, it would be worth to show also results applying other algorithms to this dense dataset. - Supplemental material is a set of disconnected sections without no reference in the main text.

Reviewer 2



Last review The paper presents a new approach for Tucker decomposition based on sketching. As classiclal Tucker Decomposition can be solved using Alternating Least Square, where each of the successive minimization problem related to one element of the decompositions (loading factors and core tensor) can be written as a matrcial LS problem, each of this successive problem can be solved approximatively using sketching. The authors propose the use of the Tensorsketch operator, defined for sketching matricial problem where some matrices have a decomposition based on kronecker product, as encountered in the successive problems of the Alternating LS. As sketching can also be used on the inner side of a matricial product, the authors describe a second algorithm where the Tensor time matrix product is sketched. Theoretical proposition on the quality of the sketched minization are also given. Last, numerical experiments on both real and synthetic data sets are presented on large datasets. In summary : the Tensorsketch operator is inserted in the Tucker-ALS algotithm which enables the decomposition of larger tensor. This is a good idea but the work itself may be too incremental as it lacks some global analysis of the algorithms presented : the impact of the sketching relatively to the global ALS scheme is not studied in theory whereas the algorithms show a relatively important relative error compared to ALS in the experiments. While it is still an alternating algorithm on the outside, the decreasing nature of ALS algorithm at each minimization is lost when the sketches are inserted, because the problem solved is not the same for each minization. How does the algorithm incurs for case when the objective function increases after solving a sketch ? The same comment can be made concerning the multipass vs one-pass. In the one pass, the sketching operators are stable, so there should be decrease between two occurrence of the same problem whereas with changing operators the problems always differs. Moreover, claims notably on streaming (in both introduction and conclusion) should be further developed as in the algorithms presented Y is a completely known input. Comments The proposition 3.1 is false in the sense that it is presented. The author pretends its conditioning is small because it is bounded by (1 +epsilon)^2/(1−epsilon)^2 for epsilon in (0,1). This is erroneous since the bound tends to infinity when epsilon tends to 1. But, the problem can be easily fixed by specifying a suitable value range for epsilon In the experiments, large but sparse tensor are generated, which can be somehow contrary to the orthogonality constraints on the factors. As the sketching do not rely on keeping the factors othogonal, a non-orthogonal version of the algorithm could be a better comparison (with maybe some Tychonov regularization of the factors)

Reviewer 3



******After rebuttal ***** The authors have addressed my minor concerns. After also seeing the other reviews, and considering the authors rebuttal, I stand by my grade of 8 and my original review. In particular, I disagree with most all of other reviewers issues and find that they are mostly non-issues. ************************** The paper uses a tensor sketching method to speed-up the computation and reduce the memory usage of Tucker decomposition methods. The authors offer two new algorithms that use the sketching, two theoretical results, and numerical experiments which are particularly convincing. I have only minor suggestions and find that the paper is at a high standard as is. The theoretical results include: Proposition 3.1 shows that one of the subproblems in their algorithm, which is a sketched linear least squares problem, has a concentrated spectrum with high probability, thus justifying their use of conjugate gradients (CG) to solve this linear least squares problem, as opposed to using a direct method. This is a welcome proposition, since indeed CG is only truly efficient at solving reasonably well conditioned problems. The second theoretical result in Proposition 3.3 is mostly indicative (though none-the-less welcome) and informally states that if a tensorsketch is re-sampled at each iteration (different from what is implemented), sketching the remaining subproblem in their sketched Tucker algorithm suffers only an additive error in comparison to the solution of the true (unsketched) linear least squares problem. Details are only given in the supplementary material. But I Their numerics comparing to several state of the art tucker decomposition technique clearly shows the superiority of their methods in the large data regime, and that the sketching only sacrifices very little in terms of the quality of the resulting decomposition. Finally, here are some minor issues and suggestions: line 69: You introduce the matrix B but later in line 71 and 72 use a matrix A. Was this intentional? line 118-119: on line 118 you give the solution to (5) which, at this point, looks impossible since it only involves adjoints of A^(i) and no pseudoinverses. But of course this follows since you explain in the next sentence that each A^(i) is given by the leading singular vectors of Z_(i). Simply re-ordering, and properly connecting these two sentences, would make it easier to follow. e.g. the solution to (5) is given by "G =..." since each A^(i) is orthogonal...etc In the supplementary material: lines 79 -83: There seems to be a mix-up between rows and columns. In particular is a_{:r_n}^(n) the r_nth column of the r_nth row of A^(n)?